

# Light use efficiency (LUE) based bimonthly gross primary productivity (GPP) for global grasslands at 30 m spatial resolution (2000–2022)

Mustafa Serkan Isik[1], Leandro Parente[1], Davide Consoli[1], Lindsey Sloat[2], Vinicius Vieira Mesquita[3], Laerte Guimaraes Ferreira[3], Simone Sabbatini[4], Radost Stanimirova[2], Nathalia Monteiro Teles[3], Nathaniel Robinson[5], Ciniro Costa Junior[6] and Tomislav Hengl[1]

[1] OpenGeoHub, Doorwerth, Netherlands
[2] Land & Carbon Lab, World Resources Institute, Washington DC, United States
[3] Remote Sensing and GIS Laboratory (LAPIG/UFG), Goiania, Brazil
[4] Euro-Mediterranean Center on Climate Change (CMCC), IAFES division, Viterbo, Italy
[5] CIFOR-ICRAF, Nairobi, Kenya
[6] Alliance of Bioversity International and CIAT, Cali, Colombia

Corresponding author
Mustafa Serkan Isik,
serkan.isik@opengeohub.org

## ABSTRACT

The article describes production of a high spatial resolution (30 m) bimonthly light use efficiency (LUE) based gross primary productivity (GPP) data set representing grasslands for the period 2000 to 2022. The data set is based on using reconstructed global complete consistent bimonthly Landsat archive (400TB of data), combined with 1 km MOD11A1 temperature data and 1° CERES Photosynthetically Active Radiation (PAR). First, the LUE model was implemented by taking the biome-specific productivity factor (maximum LUE parameter) as a global constant, producing a global bimonthly (uncalibrated) productivity data for the complete land mask. Second, the GPP 30 m bimonthly maps were derived for the global grassland annual predictions and calibrating the values based on the maximum LUE factor of $0.86\ \mathrm{gCm^{-2}d^{-1}MJ^{-1}}$. The results of validation of the produced GPP estimates based on 527 eddy covariance flux towers show an R-square between 0.48–0.71 and root mean square error (RMSE) below $\sim 2.3\ \mathrm{gCm^{-2}d^{-1}}$ for all land cover classes. Using a total of 92 flux towers located in grasslands, the validation of the GPP product calibrated for the grassland biome revealed an R-square between 0.51–0.70 and an RMSE smaller than $\sim 2\ \mathrm{gCm^{-2}d^{-1}}$. The final time-series of maps (uncalibrated and grassland GPP) are available as bimonthly (daily estimates in units of $\mathrm{gCm^{-2}d^{-1}}$) and annual (daily average accumulated by 365 days in units of $\mathrm{gCm^{-2}yr^{-1}}$) in Cloud-Optimized GeoTIFF (~23TB in size) as open data (CC-BY license). The recommended uses of data include: trend analysis *e.g.*, to determine where are the largest losses in GPP and which could be an indicator of potential land degradation, crop yield mapping and for modeling GHG fluxes at finer spatial resolution. Produced maps are available *via* SpatioTemporal Asset Catalog (http://stac.openlandmap.org) and Google Earth Engine.

# INTRODUCTION

Grasslands cover almost 40% of the Earth's land area, which makes them one of the most important ecosystems to maintain global ecological equilibrium (*Ma et al., 2024*; *Shen et al., 2015*, *2022*). Grasslands provide the necessary resources for livestock management by serving as grazing areas. They play a crucial role in supporting agriculture and ensuring food security and rural economies on a global scale (*Bengtsson et al., 2019*). Due to their extensive root systems, which allow them to encapsulate more soil organic carbon than other ecosystems, they also play an important role in the global carbon cycle. Therefore, sustainable conservation and management of grasslands, along with monitoring ecosystem health, is essential to maintain their ecological functions and to preserve their importance in reducing land degradation and climate change (*Chang et al., 2021*; *O'Mara, 2012*).

Gross primary productivity (GPP) provides important information on ecosystem health status and functionality and their role in the global carbon cycle, as well as being a measure of carbon sequestration. GPP is strongly related to the growth of vegetation, which in turn impacts species at higher trophic levels in the food chain, with consequences on the biodiversity of these ecosystems and on human economic activities (*Liu et al., 2023*). Grassland dynamics includes many factors that affect the productivity and carbon sequestration capacity of the ecosystem. Thus, GPP can be an effective indicator for understanding and assessing the role of grassland ecosystems in the carbon cycle. By modeling the spatio-temporal variations of the GPP in grasslands, we can support the conservation of natural resources, protect biodiversity and ecosystem services, and mitigate climate change (*Anav et al., 2015*; *Campbell et al., 2017*; *Hilker et al., 2008*).

High-resolution GPP modeling studies are increasingly valuable for fine-scale ecological analysis, as they capture the ecosystem responses accurately. In the last decade, several high-resolution GPP products have been developed using different methodologies and data sources, making use of recent advances in remote sensing techniques. *Robinson et al. (2018)* improved GPP and net primary productivity (NPP) products for the contiguous United States (CONUS) region using Landsat and MODIS data at fine spatial resolutions, 30 m and 250 m, respectively. In this case, the integration of finer land cover classifications and enhanced meteorological data helped to improve the existing global MODIS GPP (MOD17) product (*Running et al., 1999*). *Huang et al. (2022)* used Landsat data (2016–2020) to produce a Hi-GLASS GPP for China at 30 m spatial resolution using the Markov Chain Monte Carlo approach with Fluxnet2015 data footprints and a revised eddy covariance light use efficiency (LUE) model. Their results show that, in heterogeneous ecosystems such as wetlands, savannas, and croplands, where footprints are small and heterogeneous, the Hi-GLASS GPP product performs significantly better than coarser models (*Huang et al., 2022*). Later, this modeling framework was further advanced by incorporating a refined maize classification map and reparametrized LUE coefficients for

different ecosystem types to generate the Hi-GLASS GPP v1 dataset, offering nationwide 30 m resolution GPP estimates across China from 2016 to 2020 (*Lin et al., 2024*). Furthermore, several studies used finer spatial resolution remote sensing images in combination with downscaling approaches to model GPP at local scales (*Gitelson et al., 2012*; *Pabon-Moreno et al., 2022*; *Wolanin et al., 2019*; *Xie et al., 2023*).

Despite their limited spatial coverage, regional GPP models with higher resolution than global products provide useful insight into ecosystem dynamics by capturing small-scale variations in GPP, often missed by global models. Regardless of their significance, there are limitations to high-resolution GPP models. Because high-resolution products are computationally intensive, they generally lack global coverage. Due to the need for large amounts of storage and processing, they are also often restricted to shorter time periods. Land degradation and similar studies require the finest possible resolution data with monthly or (at least) bimonthly temporal granularity (*Hackländer et al., 2024*).

In this study, we have implemented a spacetime GPP modeling framework to map GPP for global grasslands by processing the Landsat archive (*Potapov et al., 2020*). In addition to GLAD's Landsat Analysis-Ready Data (*Potapov et al., 2020*), we also used MOD11A2 land surface temperature (*Wan, Hook & Hulley, 2021*), CERES Photosynthetically Active Radiation (PAR) (*Scott et al., 2022*) and the LUE model described in *Pei et al. (2022)* and *Chen et al. (2021)*. In addition, we generated a global intermediate product, which here we call an uncalibrated EO-based GPP (uGPP), for the entire global land mask. Here we set a constant maximum light use efficiency factor $\varepsilon_{LUE_{max}}$ of 1 $gCm^{-2}d^{-1}MJ^{-1}$, allowing data users to input their own land cover maps/classes and assign biome-specific $\varepsilon_{LUE_{max}}$ values based on more refined regional maps. To our knowledge, these global bimonthly and annual time-series maps of GPP for grasslands and uGPP for land mask are the first ever global GPP maps at 30 m spatial resolution and spanning 23 years. These datasets provide a valuable resource for ecosystem monitoring, with the calibrated GPP product specifically capturing the spatial and temporal dynamics of grasslands, while the uGPP dataset extends its applicability across all biomes. Utilizing the $\varepsilon_{LUE_{max}}$ factor specifically for grassland ecosystems, the grassland GPP product enables a precise assessment of grassland productivity, whereas uGPP offers a flexible framework to evaluate productivity in various ecosystems with unprecedented spatial resolution globally.

The source code used to produce the GPP datasets is publicly available on the GitHub page of Global Pasture Watch (https://github.com/wri/global-pasture-watch). Portions of this text were previously published as part of a preprint (https://doi.org/10.21203/rs.3.rs-5587863/v1).

## MATERIAL AND METHODOLOGY

### Overall computing framework

We developed a space-time modeling framework to produce a temporally consistent global GPP for grasslands and uGPP for land mask at 30 m spatial, which is shown in Fig. 1. This framework is implemented in a high performance computing (HPC) environment utilizing 1,400 Intel Xeon Gold CPU threads and 14 TB of RAM, and it requires approximately 48 h to complete all computations. The total size of the generated data

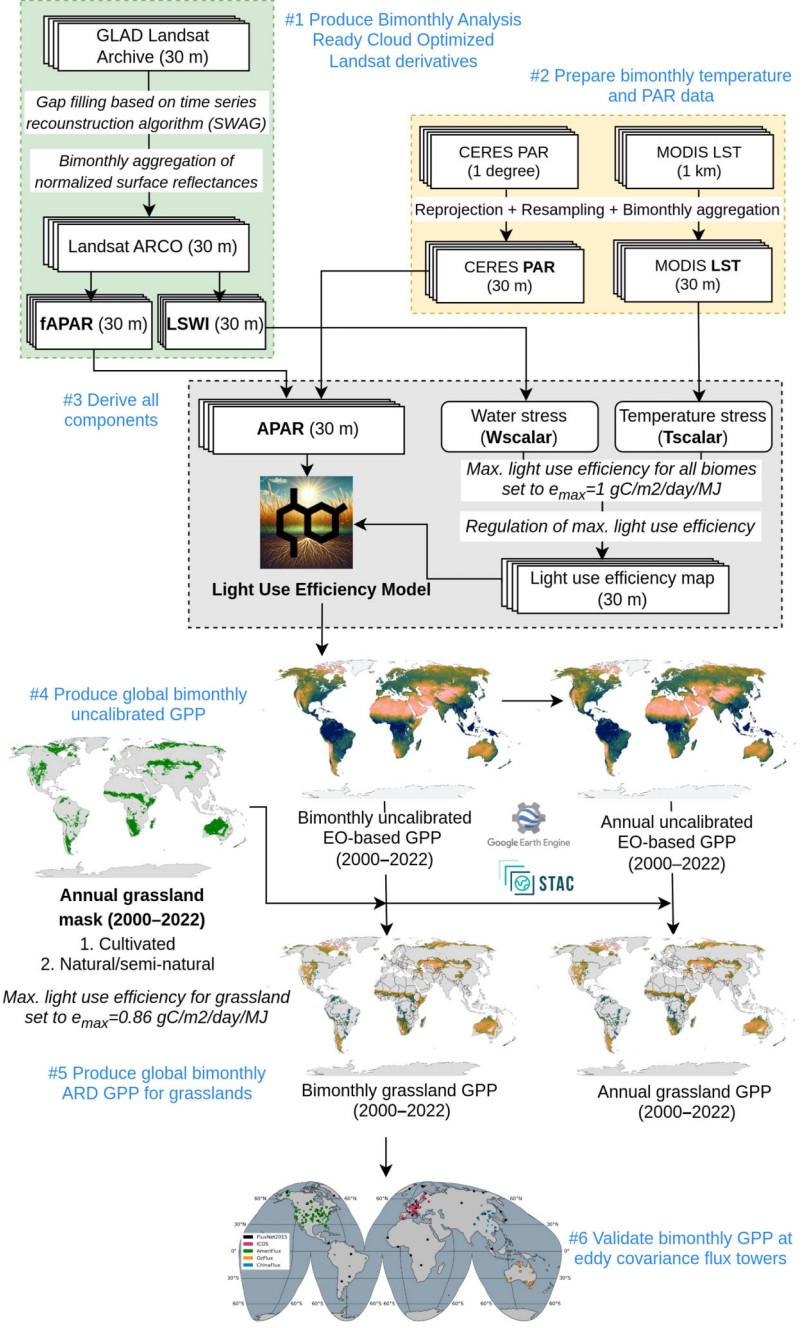

**Figure 1 Flowchart of the GPP spacetime modeling framework.**

products after compression is approximately 23 TB. The framework comprises the following steps. First, GLAD Landsat ARD version 2 (*Potapov et al., 2020*) is gap-filled using the time series reconstruction algorithm (*Consoli et al., 2024*) to produce normalized, complete and consistent bimonthly spectral reflectances. These can then be used to derive fraction of absorbed photosynthetically active radiation (FAPAR)

(*Hackländer et al., 2024*) and Land Surface Water Index (LSWI) indices bimonthly at 30 m resolution. Second, we also prepare MODIS-based land surface temperature (MOD11A2) (*Wan, Hook & Hulley, 2021*) and CERES-based PAR (*Scott et al., 2022*) layers to match exactly the same grids as Landsat images: we downscaled the 1 km resolution images to 30 m spatial resolution by cubicspline algorithm, and temporally averaged to bimonthly periods to match Landsat images. Third, all input layers are combined to produce the GPP based on the standard Light Use Efficiency model. Fourth, we derive an uncalibrated global data set for the whole mask (this is the most computational part of the procedure). Fifth, we derive an Analysis-Ready and Cloud Optimized (ARCO) GPP data set for the global map of grasslands (*Parente et al., 2024*). The final bimonthly maps are then averaged and accumulated over a 365–day period to derive annual GPP estimates.

We run this framework in OpenGeoHub's High-Performance Computing (HPC) infrastructure, where we split the GPP computation worldwide by $1° \times 1°$ tiles (about 100 km size cells) and distribute it among the processing nodes using SLURM (*Yoo, Jette & Grondona, 2003*) and Docker containers (*Boettiger, 2015*). All predicted GPP tiles were then used to create Cloud-Optimized GeoTIFF (COG) mosaics, which are publicly available in the SpatioTemporal Asset Catalog (STAC) and Google Earth Engine.

## Temperature and photosynthetically active radiation (PAR) data

To represent PAR and temperature inputs, we used CERES SYN1deg v4.1 and MOD11A2 products, respectively. CERES SYN1deg v4.1 is a global dataset that provides detailed information on the Earth's radiative energy budget, including the impact of clouds and aerosols at 1° spatial resolution (*Scott et al., 2022*). These data sets contribute significantly to a better understanding of climate dynamics and how the Earth responds to changes in atmospheric composition, especially in the context of agricultural productivity and carbon cycle studies. We used the monthly product of top-of-atmosphere radiation data to account for the radiative energy available for plants to use, the so-called PAR. The MOD11A2 product is a land surface temperature and emissivity data set based on the NASA's MODIS instrument onboard the Terra and Aqua satellites (*Wan, Hook & Hulley, 2021*). In global coverage, the MOD11A2 data set provides 8 days of average surface temperatures at a spatial resolution of 1 km since 2000. We used the daytime Land Surface Temperature (LST) to estimate a $T_{scalar}$ to limit the optimal temperature for photosynthesis. Another option to represent the monthly temperature would be, for example, the CHELSA climate time-series of interpolated air temperature (*Karger et al., 2020*).

## Landsat spectral reflectance data

We used GLAD Landsat ARD version 2 (*Potapov et al., 2020*) to produce an aggregated, cloud-free bimonthly collection of Landsat images. The original GLAD Landsat ARD is available from 1997 to 2023, however we focus on the period 2000–2022 as MODIS and other necessary layers are only available from 2000. From the GLAD Landsat we produce the so-called *"bimonthly Analysis Ready and Cloud Optimized (ARCO) cloud-free archive"* with Landsat 5–9 normalized spectral reflectance images every 2 months, which gives six

images per year or 138 images in total. Since MODIS images are only available from 2000, the first three years of the product are not used in this work. The GLAD Landsat ARD was first cloud screened using the quality assessment flag available with the data and then aggregated for every two months using four of the original 16–day images. The temporal aggregation was performed considering all clear-sky pixels for a 2–month period using a weighted average by `cloud_cover` (estimated for each date and tile).

Since after temporal aggregation a considerable amount of data was still missing (about 40% on the global scale for the entire time series), the aggregated product was imputed using the seasonally weighted average (SWAG), an algorithm that works exclusively on the temporal domain, avoiding the patching effect and loss of resolution often encountered with spatial gap-filling. In summary, absent values on each time series are reconstructed using available values, giving higher priority to pixels from the same season and relatively more recent, in order to limit erroneous land-cover change propagation. To respect causality, only images from the past of each time-stamp were used for reconstruction. More details on the data and the method used to produce bimonthly cloud-free mosaics are available in (*Consoli et al., 2024*).

## Annual maps of global grassland extent

To select grassland areas, we used a novel dataset produced by the Global Pasture Watch (GPW) initiative (https://landcarbonlab.org/data/), which maps the spatiotemporal extent of grassland from 2000 to 2022 at a 30 m spatial resolution (same temporal coverage and spatial resolution of our GPP estimates). These grassland maps were generated using 23 years of Landsat ARCO archive, together with climatic, landform, and proximity variables, 2.3 million reference samples visually interpreted using very-high resolution (VHR) imagery and spatiotemporal machine learning with Random Forest algorithm, resulting in two grassland classes (probabilities and dominant classes) mapped worldwide (*Parente et al., 2024*). The selection of the areas for mapping grassland GPP in our study considered all 30 m pixels ($\sim$900 m$^2$ as minimum detectable unit) annually mapped as cultivated and natural/semi-natural grassland, according to the dominant maps produced by GPW.

## Flux towers

To assess the performance of the GPP products, terrestrial GPP values derived from the eddy covariance flux towers were obtained as ground truth for the period 2000 to 2022. From five different data sources—FluxNet2015 (*Pastorello et al., 2020*), AmeriFlux-FLUXNET (*Novick et al., 2018*), ICOS Warm Winter 2020 (*Warm Winter 2020 Team and ICOS Ecosystem Thematic Centre, 2022*), OzFlux (*Isaac et al., 2017*), and ChinaFlux (*Yu et al., 2006*)—we extracted daily GPP values that represent productivity for a variety of biomes. The eddy covariance technique is recognized as a reliable method to directly measure the Net Ecosystem Exchange (NEE) of $CO_2$ and other trace gases at the ecosystem scale on the ground, with a robust and non-disruptive system, and their partitioned components GPP and Ecosystem Respiration (Reco). All of the selected data sources, except ChinaFlux, follow the processing approach defined in the FLUXNET context as a means of standardization, which is included in the ONEFlux processing

**Table 1 Summary of eddy covariance flux tower networks used.**

| Dataset | Flux towers All biomes | Flux towers Grassland |
|---|---|---|
| FLUXNET2015 | 206[*] | 36 |
| AmeriFlux-FLUXNET | 196[*] | 30 |
| ICOS Warm Winter 2020 | 73[**] | 9 |
| OzFlux | 28 | 7 |
| ChinaFlux | 27 | 10 |

**Notes:**
[*] 1 station was discarded due to elimination criteria.
[**] 2 stations were discarded due to elimination criteria.

package (https://github.com/fluxnet/ONEFlux; see also *Pastorello et al. (2020)* for more details). The ONEFlux package includes two different algorithms for deriving GPP estimates, the day-time (DT) and night-time (NT) approach. In addition, different realizations of GPP are obtained from a bootstrapping of different thresholds for low-turbulence conditions (see (*Papale et al., 2006*) for more details). In this study, we selected the DT-derived product, in the so-called *"reference realization"*.

As a means of ensuring the quality of the GPP data, we used daily values whose gap-filled data represents no more than 20% of the total time series. Daily data points where the GPP estimates of DT differed by more than 3 $gCm^{-2}d^{-1}$ from the estimates of NT were discarded (*Joiner et al., 2018*). As a last step, the daily estimates of the GPP were converted to bimonthly median values to match the temporal resolution of the GPP maps. The accuracy of the modeled GPP product was evaluated globally with flux towers in all biomes; however, since we primarily target grassland areas, we performed a separate assessment with stations that fall into the grassland (*GRA*) biome of the IGBP classification. The number of flux towers used for each network is summarized in Table 1.

Figure 2 shows the spatial and temporal distribution of the eddy covariance flux towers and their measurements across the networks. The global map (Fig. 2A) shows the distribution of these networks: the AmeriFlux sites are predominantly in North America and partly in South America, the ICOS sites in Europe, the OzFlux sites in Australia, and the ChinaFlux sites in China. These sites are categorized by latitudinal zones (polar, temperate and tropical) in the temporal coverage plot (Fig. 2B), which shows the operational periods from the beginning of 2000 until the end of 2022. A regional bias towards temperate ecosystems can be observed by the dense clustering of sites in the Northern Hemisphere, particularly within the temperate zone. However, the Southern Hemisphere and the tropical regions have less coverage, indicating potential under-representation of the ecosystems in these regions by observational data from flux towers, as discussed by other studies *e.g.*, *Baldocchi, Chu & Reichstein (2018)*.

## GPP estimation based on light use efficiency model

The LUE models estimate the productivity in terrestrial ecosystems based on the amount of energy absorbed from the available solar radiation by plants and converted to carbon *via* photosynthetic activities (*Monteith, 1972*). Assuming a linear relation between the

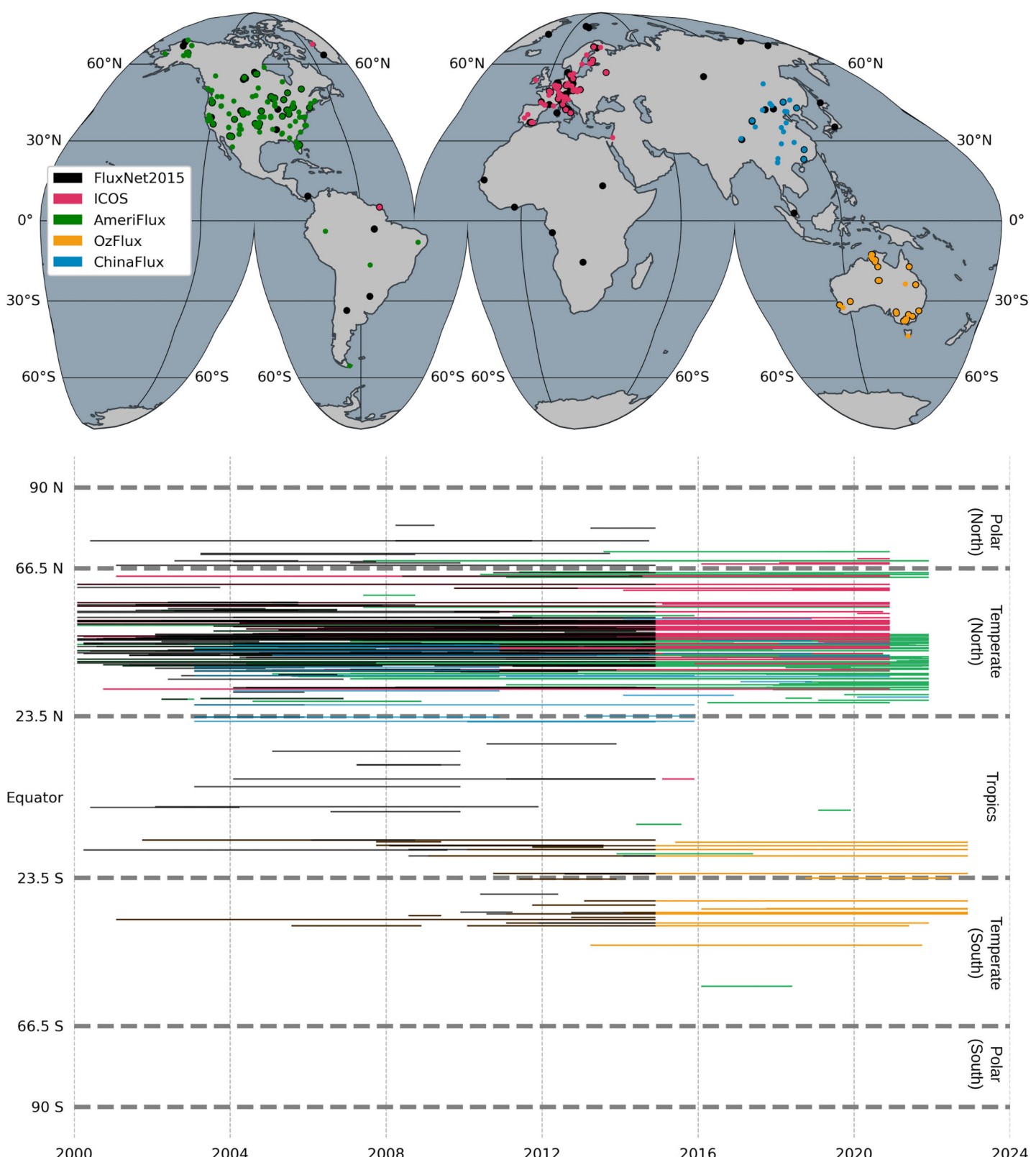

**Figure 2 The locations of flux towers (top) and the temporal coverage of the GPP time series in latitudinal zones (bottom).**

productivity and the absorbed photosynthetically active radiation (APAR), the LUE model defines GPP in a simple manner as follows:

$$GPP = PAR \times fAPAR \times \varepsilon_{LUE} \qquad (1)$$

where PAR is photosynthetically active radiation calculated from short-wave radiation, fAPAR is the fraction of PAR absorbed by the canopy and $\varepsilon$ is the light use efficiency (LUE) coefficient (*Pei et al., 2022*). The computation of fAPAR is based on its relation to NDVI values and it is formulated as:

$$fAPAR = \frac{(NDVI - NDVI_{min}) \times (fAPAR_{max} - fAPAR_{min})}{NDVI_{max} - NDVI_{min}} + fAPAR_{min} \qquad (2)$$

where, $NDVI_{min}$ and $NDVI_{max}$ correspond to the 2% and 98% threshold values of NDVI distribution and are 0.03 and 0.96, respectively. $fAPAR_{min}$ and $fAPAR_{max}$ are the theoretical minimum and maximum FPAR for all vegetations and are estimated as 0.001 and 0.95 (*Robinson et al., 2018*; *Wang et al., 2014*).

In LUE-based modeling, the conversion of APAR to carbon is determined by the factor of light use efficiency $\varepsilon_{LUE}$. During this conversion, the model relies on the physiological principle that the vegetation converts APAR into biochemical energy through photosynthesis by operating at their maximum potential efficiency. However, there are environmental factors that adversely affect the efficiency of the energy converted, particularly temperature and water availability, which are the most limiting factors. By adjusting the theoretical maximum value of the LUE factor ($\varepsilon_{LUE_{max}}$) with temperature scalar ($T_{scalar}$) and water condition of the land surface ($W_{scalar}$), the model takes into account these environmental constraints on the productivity of vegetation. The relation between $\varepsilon_{LUE}$ and these scalars can be formulated as *Jin et al. (2013)*:

$$\varepsilon_{LUE} = \varepsilon_{LUE_{max}} \times T_{scalar} \times W_{scalar} \qquad (3)$$

where $T_{scalar}$ affects the rate of biochemical processes in plants. It can be formulated as:

$$T_{scalar} = \frac{(T - T_{max}) \times (T - T_{min})}{(T - T_{max}) \times (T - T_{min}) - (T - T_{opt})^2} \qquad (4)$$

where $T_{min}$, $T_{max}$, and $T_{opt}$ are the minimum, maximum, and optimal temperatures for photosynthesis in units Celsius, and are taken as 0.0 °C, 48.0 °C, and 20.3 °C, respectively (*Yuan et al., 2007*). $T$ represents the mean temperature over a period of time, which is bimonthly for this study. When zero or negative temperatures are introduced into the equation, the $T_{scalar}$ value is limited to zero (*Raich et al., 1991*).

The water content in the soil is represented by $W_{scalar}$ parameter which reflects the availability of the water content in the soil to indicate the level of water stress. It can be expressed as a function of land surface water index (LSWI) (*Robinson et al., 2018*):

$$W_{scalar} = 1 - \frac{1 - LSWI}{1 + LSWI_{max}} \qquad (5)$$

where LSWI is calculated using the near infrared (NIR) and short-wave infrared (SWIR) bands using Eq. (6), and $LSWI_{max}$ represents the maximum LSWI value in a bimonthly period.

$$LSWI = \frac{NIR - SWIR}{NIR + SWIR}. \tag{6}$$

The $\varepsilon_{LUE_{max}}$ factor in Eq. (3) determines the amount of energy that plants can use in a given area under optimal conditions and is determined by factors such as the availability of light, the temperature and the conditions of the water of plants. $\varepsilon_{LUE_{max}}$ is adjusted according to land cover and climatic factors, which makes it a biome-specific physiological parameter. Since it theoretically represents the maximum conversion rate of vegetation, its relation to GPP is subdued by the environmental stress factors, given in Eqs. (4), (5) that affect vegetation productivity. Hence, the LUE factor $\varepsilon$, in Eq. (3), becomes the actual LUE calculated from its theoretical value $\varepsilon_{LUE_{max}}$ by regulating it with the environmental conditions that cause biophysical and biochemical stress in vegetation, such as temperature and water stress. In fact, it can be estimated for different ecosystems to account for the heterogeneity of productivity between regions under varying climatic conditions (*Madani et al., 2014*; *Sánchez et al., 2015*; *Wang et al., 2010*).

## Spatio-temporal modeling of GPP

As part of a computational framework that we used to model spatio-temporal variations in GPP at 30 m resolution, we produced a bimonthly version of the Landsat ARCO archive in $1° \times 1°$ tiles between 2000 and 2022. Using the red, NIR, and SWIR-1 bands, fAPAR (Eq. (2)) and LSWI (Eq. (6)) we computed a time-series of images for the whole period of interest. We averaged the time series of MODIS-based LST and CERES-based PAR estimates to produce bimonthly mean values for the same period and interpolated the pixel values with cubic splines to match the 30 m spatial resolution of the Landsat archive. The LUE model was applied to produce GPP estimates for the same $1° \times 1°$ tiles used in the GLAD products. The GPP productions for the tiles were carried out using an open source Python library (scikit-map), which was created by *Consoli et al. (2024)*. The tiles were later mosaicked to produce COG files. Using Docker containers and SLURM, the computation process, from the implementation of the LUE model to the creation of final global mosaics, was distributed among a large number of processing nodes. This was done in a HPC environment.

We developed the GPP model for global grassland regions in two steps. As a first step, we implemented the light use efficiency model in Eq. (1) globally and produced a GPP map independent of any land cover product. Instead of applying a biome-specific productivity factor, we set the maximum light use efficiency parameter ($\varepsilon_{LUE_{max}}$) to 1 $gCm^{-2}d^{-1}MJ^{-1}$ regardless of the biome type to produce land cover independent productivity (uGPP) maps. By doing so, it is possible to calibrate GPP values using any land cover map, as well as to adjust light use efficiency factors according to local/regional dynamics.

In the second step, we produced the GPP maps for the grassland biome by masking the uGPP maps with the global grassland maps produced between 2000–2022

(*Parente et al., 2024*). We further processed the annual dominant class maps, produced using the probabilities of grassland classes, by setting the maximum spatial extent of the cultivated and natural/semi-natural maps of grasslands for each year to generate annual grassland masks. The uGPP values of these masked grassland regions are calibrated using the $\varepsilon_{LUE_{max}}$ factor (0.86 $gCm^{-2}d^{-1}MJ^{-1}$) of grassland biomes according to the global adjusted estimation in the Biome Property Lookup Table in the MOD17 algorithm (*Running et al., 1999*). It is important to note that $\varepsilon_{LUE_{max}}$ is not applied directly to uGPP derivation, instead it is set as a scale factor in the metadata of COG files, along with the scale factor of 0.1 that accounts for converting GPP values from integer to floating values (scale = 0.086). The maps represent the GPP estimates in units of $gCm^{-2}d^{-1}$, which means that the GPP values are daily estimates of productivity in a bimonthly period.

In addition to deriving bimonthly uGPP and grassland GPP products, we generated annual GPP products during the same time period. The annual GPP values were calculated by averaging the daily GPP estimates of all bimonthly periods within each year and accumulating the average value by multiplying it by 365 days to derive the annual estimate in units of $gCm^{-2}yr^{-1}$.

## RESULTS

### Production of global GPP maps

Figure 3 illustrates the bimonthly uGPP map for the year 2022 in units of $gCm^{-2}d^{-1}$, which reveals the spatio-temporal distribution of carbon uptake as a result of photosynthetic activity on a global scale. These maps represent the daily value of a gram of carbon for the corresponding bimonthly period under the assumption that fAPAR is converted into carbon at the same rate across all vegetation biomes. It can be seen from the maps that there are seasonal patterns in the levels of productivity, with high GPP values (green) seen in tropical and temperate regions during peak growth periods, and lower GPP values (brown) observed during the colder months in the arid and boreal regions. It is important to note that these uGPP maps cannot be used directly in analysis where higher accuracy is needed without calibrating the GPP values based on the maximum light use efficiency factors specific to any biome, using land cover information.

Following bimonthly uGPP derivation, grassland GPP products are produced by masking uGPP maps with the spatial extent of annual grassland masks between 2000 and 2022. The uGPP values of these masked grassland pixels are calibrated using the $\varepsilon_{LUE_{max}}$ factor (0.86 $gCm^{-2}d^{-1}MJ^{-1}$) of the grassland biome according to the globally adjusted estimation in the biome property lookup table in the MOD17 algorithm (see Table 2).

Annual GPP maps provide a measure of the total carbon dioxide captured by vegetation through photosynthesis over a year. To create these maps, we calculate the annual uGPP and grassland GPP by averaging the bimonthly maps for each year and then aggregating the values to cover the full 365-day period, expressed in units of $gCm^{-2}yr^{-1}$ (see Fig. 4).

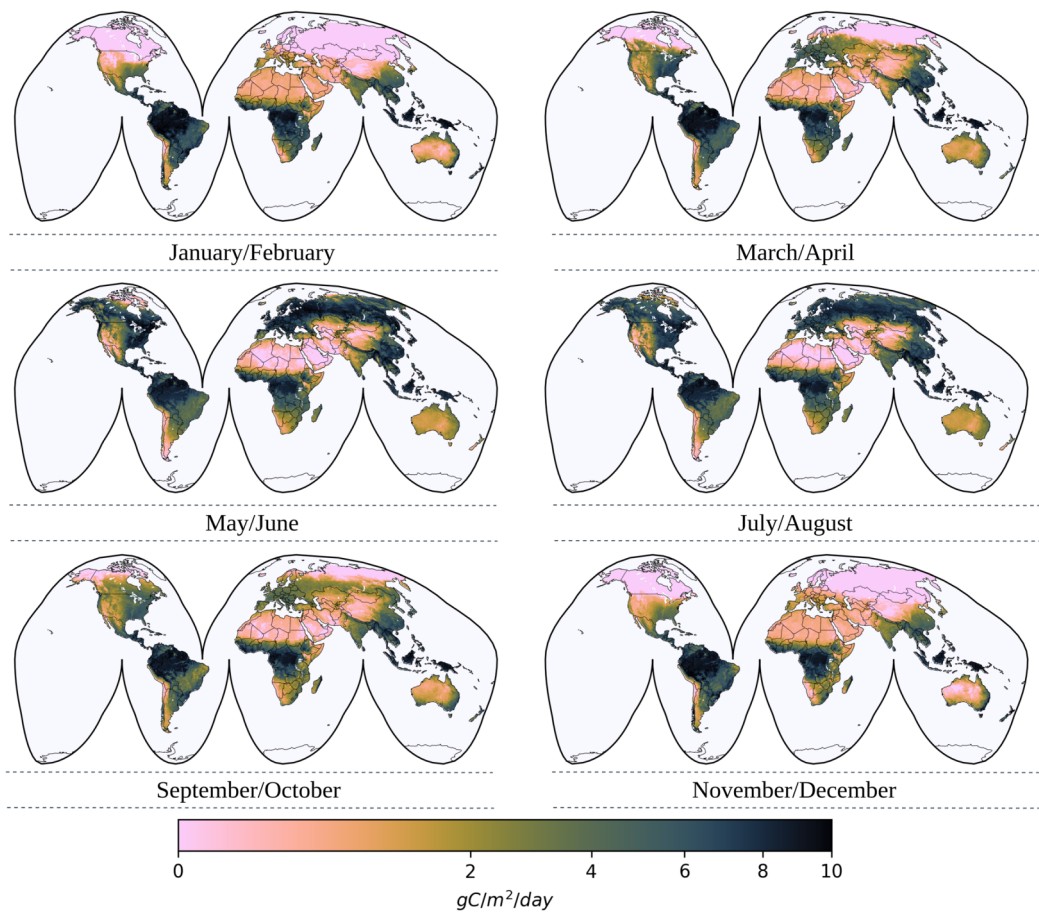

**Figure 3 Global bimonthly maps of uGPP for 2022.** Each bimonthly map shows spatial variations in the daily estimation of uGPP where low GPP values are represented in brown and high GPP values are in green.

**Table 2 Maximum light use efficiency factor in gCm⁻²d⁻¹MJ⁻¹ unit for different biomes according to IGBP land cover classification (*Running et al., 1999*).**

| Class name | Class code | $\varepsilon_{\mathrm{LUE_{max}}}$ |
|---|---|---|
| Evergreen Needleleaf Forests | ENF | 0.962 |
| Evergreen Broadleaf Forests | EBF | 1.268 |
| Deciduous Needleleaf Forests | DNF | 1.086 |
| Deciduous Broadleaf Forests | DBF | 1.165 |
| Mixed Forests | MF | 1.051 |
| Closed Shrublands | CSH | 1.281 |
| Open Shrublands | OSH | 0.841 |
| Woody Savannas | WSA | 1.239 |
| Savannas | SAV | 1.206 |
| Grasslands | GRA | 0.860 |
| Croplands | CRO | 1.044 |
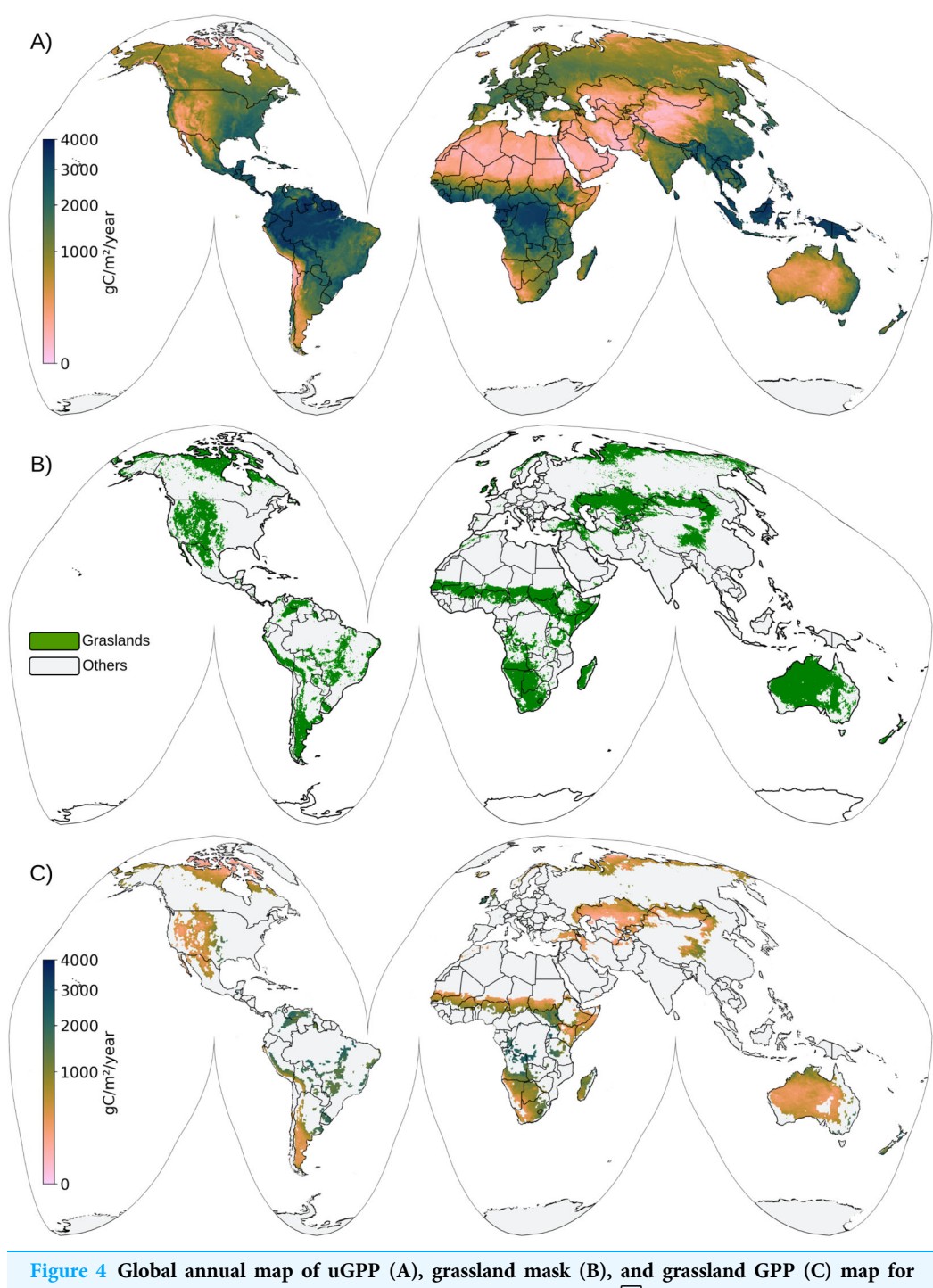

**Figure 4 Global annual map of uGPP (A), grassland mask (B), and grassland GPP (C) map for 2021.**

## Accessing the GPP datasets

Bimonthly and annual uGPP and grassland GPP maps can be publicly accessed through OpenLandMap STAC (SpatioTemporal Asset Catalog). The COG format and S3-based cloud storage of these products simplify access to large amounts of raster data and enable

fast retrieval and visual inspection of these layers by *e.g.*, QGIS (https://www.qgis.org/) and programing languages (Python computational notebooks). Below are the STAC entries for accessing all data produced by this study:

- **uGPP:**

  - Bimonthly: https://stac.openlandmap.org/gpw_ugpp.daily-30m/collection.json
  - Annual: https://stac.openlandmap.org/gpw_ugpp-30m/collection.json

- **Grassland GPP:**

  - Bimonthly: https://stac.openlandmap.org/gpw_gpp.daily.grass-30m/collection.json
  - Annual: https://stac.openlandmap.org/gpw_gpp.grass-30m/collection.json

Due to the large file sizes, we recommend that users avoid downloading the entire file at once by copying and pasting the link directly into a web browser or similar. Instead, it is recommended to use the link within Geographic Information System (GIS) software, such as QGIS, to access and work with the data layers. Using GIS software allows users to crop the data to a specific region of interest. They can also utilize a programming language to handle the data in the STAC catalog efficiently.

In addition to STAC, annual uGPP and grassland GPP maps can be accessed through the GEE data catalog (https://developers.google.com/earth-engine/datasets/publisher/global-pasture-watch). These annual layers can be accessed by their image collection ID, which are as follows:

- **uGPP:** projects/global-pasture-watch/assets/ggpp-30m/v1/ugpp_m
- **Grassland GPP:** projects/global-pasture-watch/assets/ggpp-30m/v1/gpp-grassland_m

## Validation at flux towers

In order to assess the accuracy of GPP estimations, the modeled GPP values were spatio-temporally aligned with the eddy covariance flux tower measurements integrated for every bimonthly period from 2000 to 2022. GPP estimates of the model that correspond to each bimonthly value of the flux towers were calculated by approximating the flux tower footprints with a circle of 250 m radius, which was determined empirically. The performance of the GPP estimates was assessed based on Pearson's correlation coefficient (r), root mean square error (RMSE), bias, and the coefficient of determination ($R^2$).

The validation of the GPP product is divided into two parts. The first part of the validation assessed the overall performance of the uGPP estimates across all types of land cover to demonstrate its applicability on a global scale in varying climatic regions and over a variety of types of land cover. Before comparison, uGPP estimations were converted to final GPP values, comparable with GPP measurements at flux towers, by applying the $\varepsilon_{LUE_{max}}$ factor from the MOD17 lookup table with the land cover information of the flux towers in IGBP classification (see Table 2). Later, we focused specifically on grassland

biomes worldwide to evaluate the product's accuracy within these ecosystems, since that was the main goal of this article.

Figure 5 presents the validation statistics of the GPP product across different flux tower networks for all land cover types (All) and specifically for grassland sites (GRA). The figure combines scatter plots of the GPP estimates from the LUE model and *in-situ* observations from flux networks, and accuracy statistics, with the left column showing results for all land cover types and the right column focusing on grassland sites. Each row in the figure represents datasets from different networks. In general, our model performs well on a variety of ecosystems, as depicted by the statistics for all types of land cover. Compared to the ICOS network, the model performed well for all types of land cover, explaining 71% of the variability in GPP ($R^2 = 0.71$), having a significant correlation (0.85), and the lowest RMSE (1.94 $gCm^{-2}d^{-1}$). This may indicate that the model performs best in climate regions with mild winters and moderate temperatures, as the ICOS Warm Winter dataset represents. The model shows good performance when compared to Fluxnet2015 data, with a high correlation (0.79) and slightly lower $R^2$ value ($R^2 = 0.58$) compared to the ICOS network, but is still indicative of model's accuracy across a wide range of ecosystem types. The performance of the model was lower compared to the Ameriflux, Ozflux, and ChinaFlux networks with $R^2$ values near $\sim 0.5$ and slightly higher RMSE values. Nevertheless, the GPP estimations are highly correlated with the observations from all networks with Pearsons' correlation coefficients ranging between 0.72 and 0.85.

Similarly, the GPP model demonstrated a high level of accuracy in estimating productivity for grassland ecosystems across all networks, with correlation values ranging between 0.73 and 0.90, and RMSE values below $\sim 2$ $gCm^{-2}d^{-1}$. For the Fluxnet2015 network, the model explained 70% of the variability in GPP observed by the flux towers. Compared to the ICOS Warm Winter data set, the model is in strong agreement with the ICOS Warm Winter data set, showing the highest correlation (0.90) and $R^2$ of 0.66. However, it showed less agreement with the grassland ecosystems covered by Ameriflux, Ozflux, and Chinaflux networks, as indicated by the $R^2$ values dropping to $0.51 - 0.54$, despite the high correlation coefficients between $0.73 - 0.84$. One notable trend in grassland comparisons is the consistency of the negative bias observed for networks. This negative trend might indicate a tendency for our model to potentially underestimate GPP values in grassland regions covered by these networks.

Generally, the GPP estimates from the LUE model tend to fall along the 1:1 line compared to all networks, which is also supported by the high correlation values observed against *in-situ* measurements. However, on the left side of the figure where the GPP estimates for all land covers are shown together, we can observe the underestimation problem of the LUE model, specifically at high GPP values in Fluxnet2015, AmeriFlux, and ICOS Warm Winter datasets, and notably less in Chinaflux dataset (Figs. 5A, 5C, 5E, 5I). Most of these underestimations are observed in cropland regions. Contrary to these estimations, the model experiences a slight overestimation problem in the Ozflux network (Fig. 5G). The model estimations that are above $\sim 15$ $gCm^{-2}d^{-1}$ are in fact observed between 5 and 10 $gCm^{-2}d^{-1}$ at the flux towers. These observations, as they commonly exist in both datasets, are visible in the scatter plot for Fluxnet2015 as well. However, the

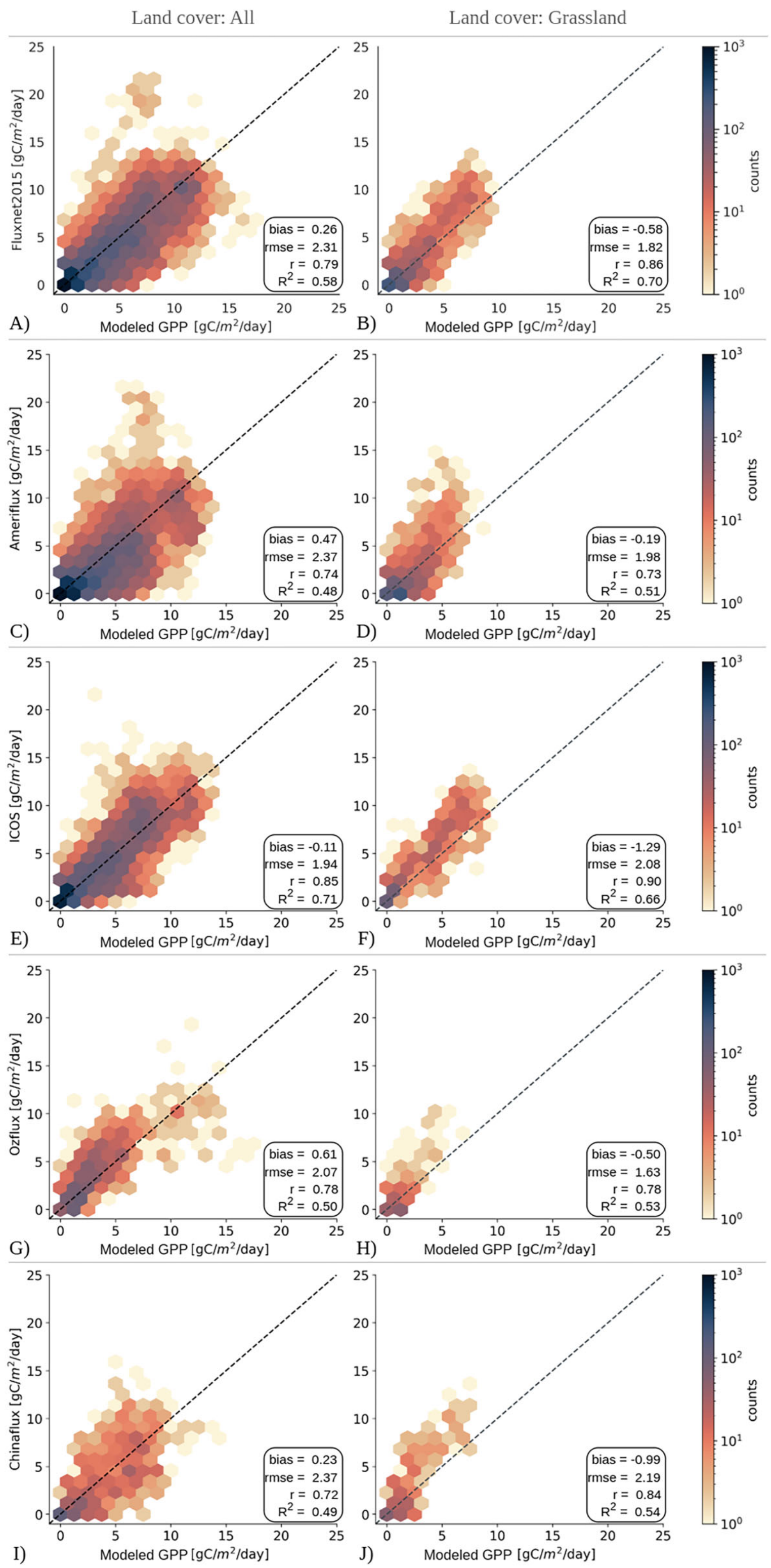

**Figure 5 Evaluation of modeled GPP for all land covers and grasslands at flux towers from (A, B) FLUXNET2015, (C, D) AmeriFlux-FLUXNET, (E, F) ICOS Warm Winter 2020, (G, H) OzFlux, and (I, J) ChinaFlux network.** Dashed lines show the 1:1 lines.

grassland comparison illustrates how well the LUE model can capture the GPP variations at the flux towers. In this case, the underestimations of GPP values are lessened, yet still remain an issue for the LUE model. These underestimation patterns in grasslands and, by extension, in other land covers are possible indications for variations in environmental factors or unique site attributes that cannot be completely accounted for by the model.

Figure 6 shows seasonal and interannual variations for GPP, temperature, and precipitation at four flux towers in grassland biomes. Each time series shows the observed GPP and the modeled GPP values together with the temperature (°C) and precipitation (mm) patterns in the bimonthly periods, to visualize how environmental factors influence the dynamic of carbon fluctuations. Strong seasonal cycles frequently appear in both observed and modeled GPP, having their peak during the warmer months that correlate with higher temperatures and, occasionally, precipitation. The temperature sensitivity of GPP is demonstrated by aligning GPP peaks at higher temperatures in temperate regions such as Horstermeer (NL-Hor), Grillenburg (DE-Gri), and California (US-Var). Riggs Creek (AU-Rig), Australia, on the other hand, has a more fluctuating pattern with irregular precipitation and lower GPP amplitudes, suggesting possible water constraints in the surrounding biome. There is a close match between observed and modeled GPP across different sites, demonstrating that the model can capture seasonal GPP trends, but some discrepancies suggest possible model refinements, particularly in modeling the peaks of inter-annual variability in arid and semi-arid regions where precipitation plays a much greater role in GPP variability.

## Comparison with other GPP products

There are well-established GPP products available globally that adopt different modeling strategies or environmental parameters. These variations in the modeling strategies can lead to distinct spatio-temporal patterns between these data products, especially if the spatial resolutions vary. It is important to be aware of the differences between GPP estimations and the potential biases associated with data products. In this context, we evaluated the performance of our GPP estimates by comparing them with two benchmark datasets, MOD17 GPP (*Running et al., 1999*) and PML (Penman-Monteith-Leuning) GPP (*Zhang et al., 2019*) models, across different regions where grasslands dominate the landscape. Figure 7 illustrates the comparison of GPP estimates for two regions of interest in South America and Southern Africa, shown as Regions A and B on the map above, together with high-resolution Google satellite imagery as a reference for the details of the landscape. For each region with 3° × 3° extent, GPP estimates from three sources, our model (30 m), PML (500 m), and MODIS (500 m) are compared visually in the first row. In the second row of the comparisons, a smaller sub-region is highlighted with a red box in all maps in the first row, and plotted in the second row with a larger map scale to be able to clearly show the impact of high-resolution GPP estimations compared to coarser resolution of the other two models. We removed all GPP values outside the grassland regions in all GPP products using grassland masks. Our GPP maps in both Region A (Southern Africa) and Region B (South America) show fine-scale changes in productivity, with detailed spatial patterns influenced by landforms and vegetation cover. However, the

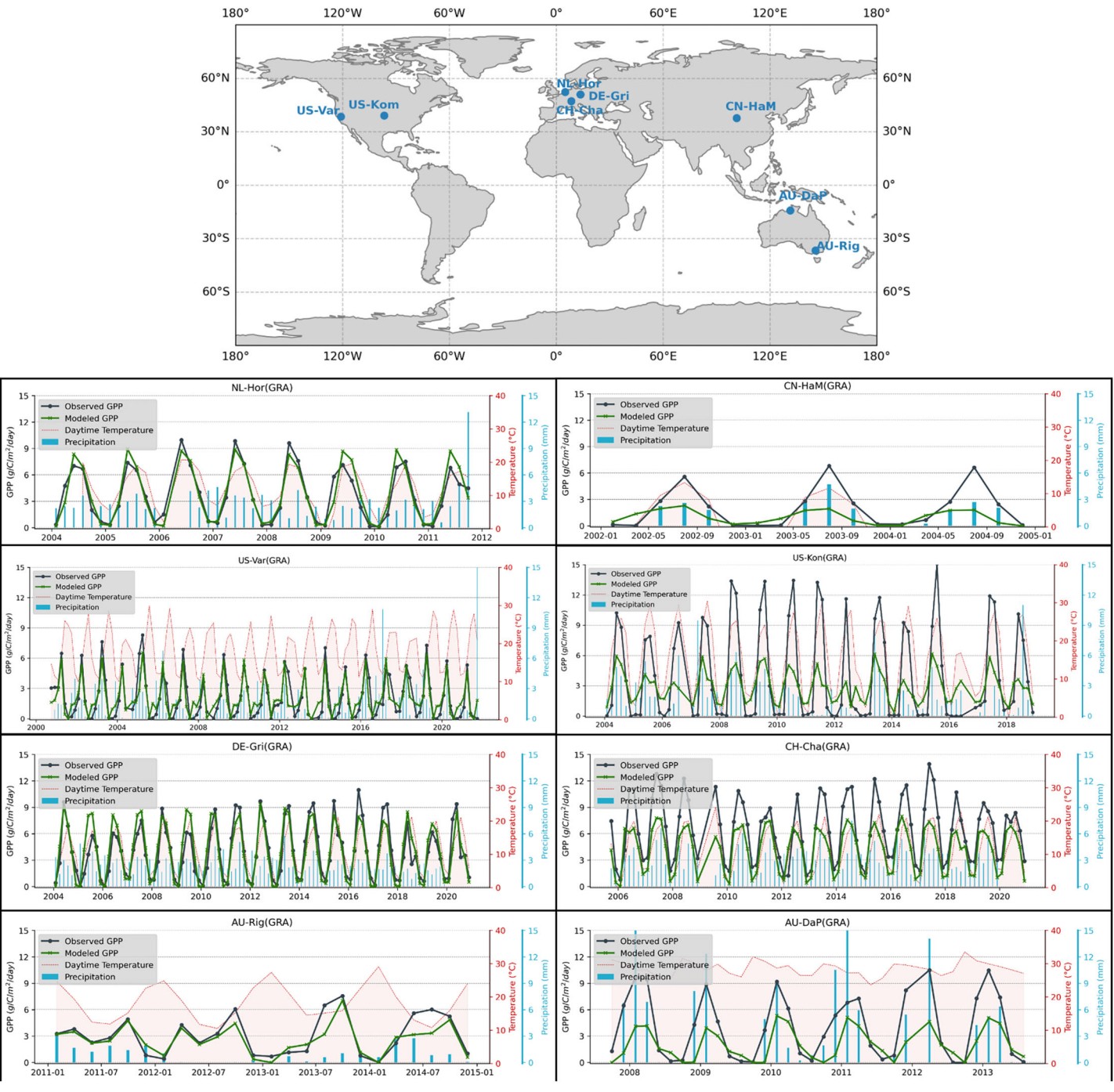

**Figure 6 Examples of long-term measurements of flux towers in grasslands for representative biomes and comparison with our modeled GPP.** From first row to the last, the stations are selected from Fluxnet2015, Ameriflux, ICOS Warm Winter 2020, and Ozflux, respectively.

PML and MODIS maps, both of which have a 500 m resolution, can only show the broader spatial patterns of the GPP. The finest features of the landscape that are easily spotted on our map are noticeably smoothed down. Although the PML and MODIS maps cannot
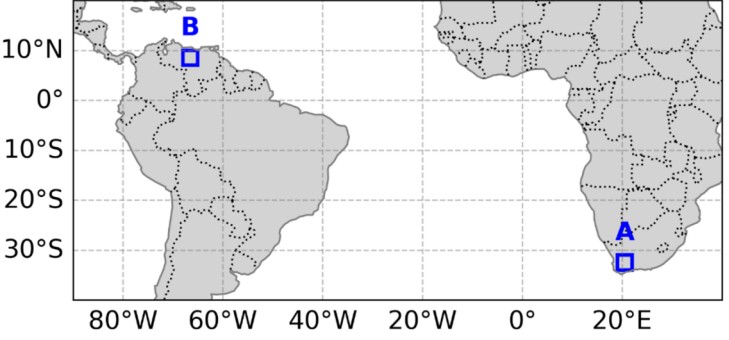

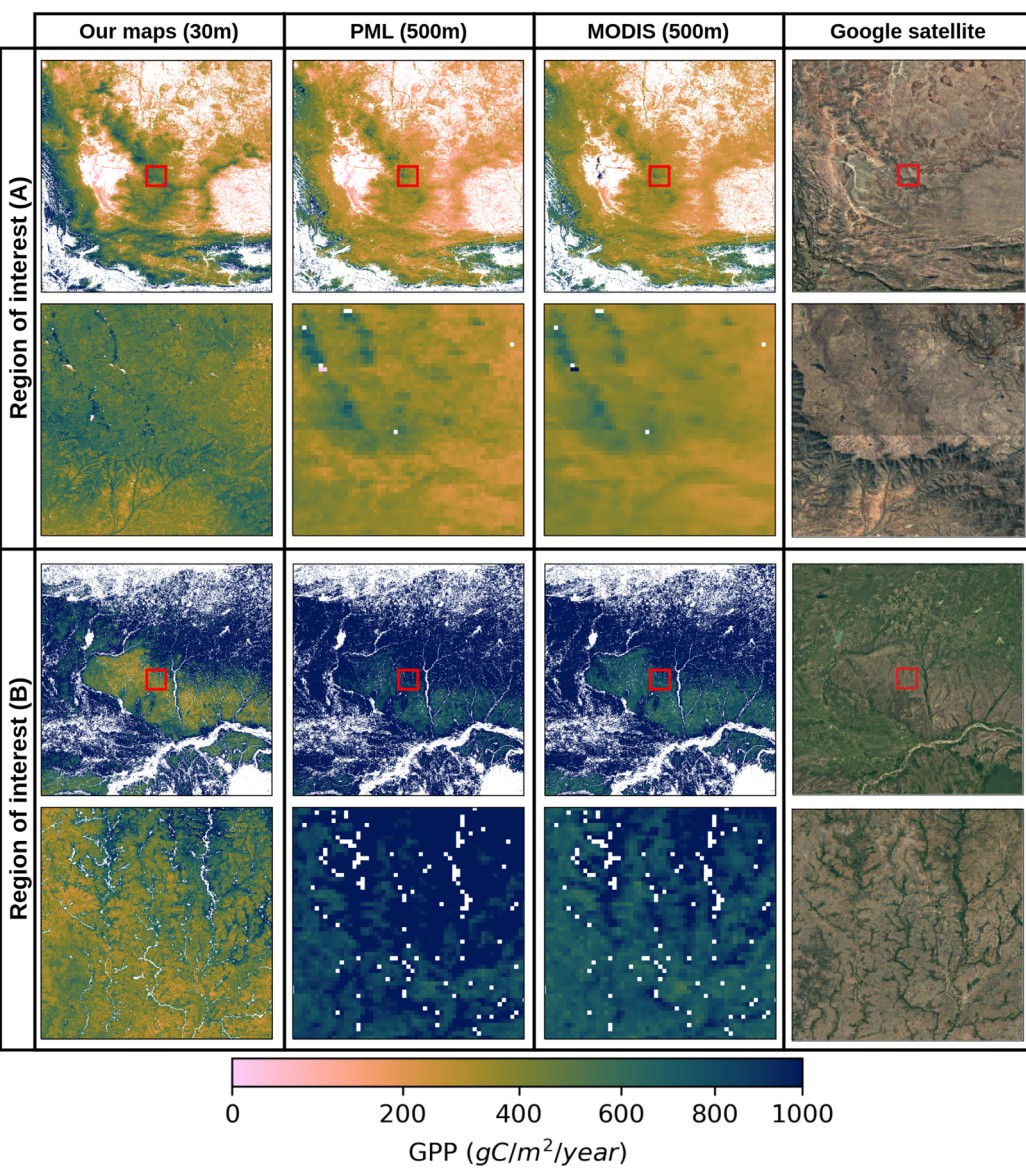

**Figure 7** Comparison of annual GPP estimations in grasslands over the South of Africa in 2021 and South America in 2020, using three different GPP products at varying spatial resolutions: our map at 30 m, PML (500 m), and MODIS (500 m). Red rectangles in the center of each GPP map correspond to 0.25° × 0.25° close-up view in that region to emphasize the different resolutions. Satellite image: © 2025 Google.

depict the subtle variation observed at the 30 m resolution, they show comparable large-scale patterns, clearly identifying regions of higher and lower productivity.

It should be noted that our model estimates a lower amount of annual GPP values in the center of the region in South America and slightly higher values in South Africa, compared to the PML and MODIS products. This is even more visible in the close-up view in that region, which may stem from variations in the input data or modeling strategies. Considering the similarities between MOD17 and our model in terms of how LUE is implemented, despite small differences in the estimations of temperature and water scalars, some of these variations can arise from differences in land cover inconsistencies. More importantly, the high temporal resolution of the MODIS and PML models can lead to a more accurate representation of carbon stocks annually, as they allow catching intraseasonal variations in highly productive ecosystems. Our model, on the other hand, is sensitive to the changes in productivity in a bimonthly frame and thus may lead to underestimations in such regions.

As one of the publicly available GPP products, the models produced (*Robinson et al., 2018*) serve as a valuable data set to understand the productivity of the ecosystem in the CONUS region. In *Robinson et al. (2018)*, two sets of GPP products based on the LUE model are presented: 30 m resolution GPP using Landsat and 250 m resolution GPP using MODIS over the CONUS region, where the $\varepsilon_{LUE_{max}}$ factors for different land cover types in the MOD17 algorithm were optimized using flux towers. To compare the performance of our GPP product and these GPP models, we geographically restricted the FLUXNET2015 data to the CONUS region. Table 3 summarizes the validation statistics of GPP models compared to eddy covariance flux tower measurements over the CONUS region. In particular, optimized $GPP_{M250}$ and $GPP_{L30}$ models show substantial improvements over their MOD17 counterparts. Our model demonstrates strong performance, especially for grasslands, achieving the highest Pearson's correlation (0.78) and the lowest RMSE (1.53) bias (−0.05), showing that it captures grassland GPP dynamics very well.

In order to visually assess the performance of different GPP products in the CONUS region and show the impact of spatial resolution across different landscapes, we selected two regions of interest where the grassland biome dominates the majority of the pixels. Figure 8 shows the visual assessment of the GPP maps in these selected regions (region A and region B) to reveal the strengths and limitations of using varying spatial resolutions as well as a variety of data sources and modeling strategies. This comparison clearly shows the advantages of high-resolution GPP estimations to observe detailed representations of spatial variations in grassland ecosystems. The coarser MODIS based products (250 and 500 m) and PML model (500 m), on the other hand, provide a more generalized view of GPP patterns, which makes it difficult to understand the small-scale variations of grassland dynamics. Although spatial variations in the GPP based on landscape features are more visible in 250 m resolution, these coarse resolution maps are useful to monitor regional dynamics from a broader perspective.

The spatial variability of grassland productivity in its finest details can be clearly observed in our map and the Landsat-based GPP product ($GPP_{L30}$). In general view where regions A and B are shown within the $3° × 3°$ plots, both maps can clearly show the broad

**Table 3 Statistics of GPP validation at eddy covariance flux towers over the CONUS region.** The best results are highlighted in bold.

| Land cover | Model | Pearson's r | RMSE | Bias |
|---|---|---|---|---|
| All | $GPP_{M250}$ (MOD17) | 0.60 | 4.33 | 1.90 |
| | $GPP_{M250}$ (Optimized) | 0.79 | 2.83 | **0.02** |
| | $GPP_{L30}$ (MOD17) | 0.63 | 4.13 | 1.66 |
| | $GPP_{L30}$ (Optimized) | **0.80** | 2.83 | 0.08 |
| | Our model | 0.74 | **2.53** | 0.43 |
| GRA | $GPP_{M250}$ (MOD17) | 0.71 | 2.52 | 1.38 |
| | $GPP_{M250}$ (Optimized) | 0.75 | 1.85 | −0.15 |
| | $GPP_{L30}$ (MOD17) | 0.69 | 2.29 | 1.00 |
| | $GPP_{L30}$ (Optimized) | 0.72 | 2.01 | 0.39 |
| | Our model | **0.78** | **1.53** | **−0.05** |

spatial pattern of the GPP in strong agreement and capture the main productivity gradients, despite minor differences. However, in the zoomed images, the maps reveal some noticeable variations, specifically in region B. One of the important sources of these variations is the grassland mask that we applied to GPP products. In region B, the high GPP values (green colors) dominated in $GPP_{L30}$ product are, in fact, classified as cultivated croplands in the annual land cover map used by *Robinson et al. (2018)*, but they are labeled grasslands on our map. Since we calibrated the productivity in grassland regions using the maximum light use efficiency factor of $\varepsilon_{LUE_{max}} = 0.86$ gCm$^{-2}$d$^{-1}$MJ$^{-1}$ from the MOD17 lookup table (see Table 2), and the corresponding pixels in $GPP_{L30}$ product are calibrated with a factor of $\varepsilon_{LUE_{max}} = 1.76$ gCm$^{-2}$d$^{-1}$MJ$^{-1}$ for cropland regions (see *Robinson et al., 2018*: Table 2), these pixels have lower GPP values in our maps. The same problem is apparent compared to the MODIS-based GPP product ($GPP_{M250}$) for which the calibration factor in the croplands is determined even higher ($\varepsilon_{LUE_{max}} = 2.27$ gCm$^{-2}$d$^{-1}$MJ$^{-1}$). Furthermore, our grassland map applies a broader definition to include pasture lands, which leads to an extensive classification of grasslands that also encompass shrubland regions. But these two classes are distinctly separated in the National Land Cover Database annual maps (*Yang et al., 2018*) which results in a narrower representation of grasslands extent. Since the maximum light use efficiency factors for shrubland and grassland are different, this causes variations in GPP estimates between different products.

Similar to the high resolution GPP product for the CONUS region, Hi-GLASS GPP v1 (*Lin et al., 2024*) represents a significant improvement in public gross primary productivity (GPP) products, offering nationwide coverage of China at fine spatial resolutions of 30 m. The model relies on Landsat imagery collected between 2016 and 2020 and adopts a crop and ecosystem specific LUE modeling framework to estimate productivity. The LUE model incorporates eddy covariance flux tower data to calibrate $\varepsilon_{LUE_{max}}$, thereby improving the representation of spatial and temporal variability in GPP over heterogeneous landscapes.

To illustrate differences in spatial patterns and magnitudes among GPP products, we presented a visual comparison of our GPP estimations over two regions in China using

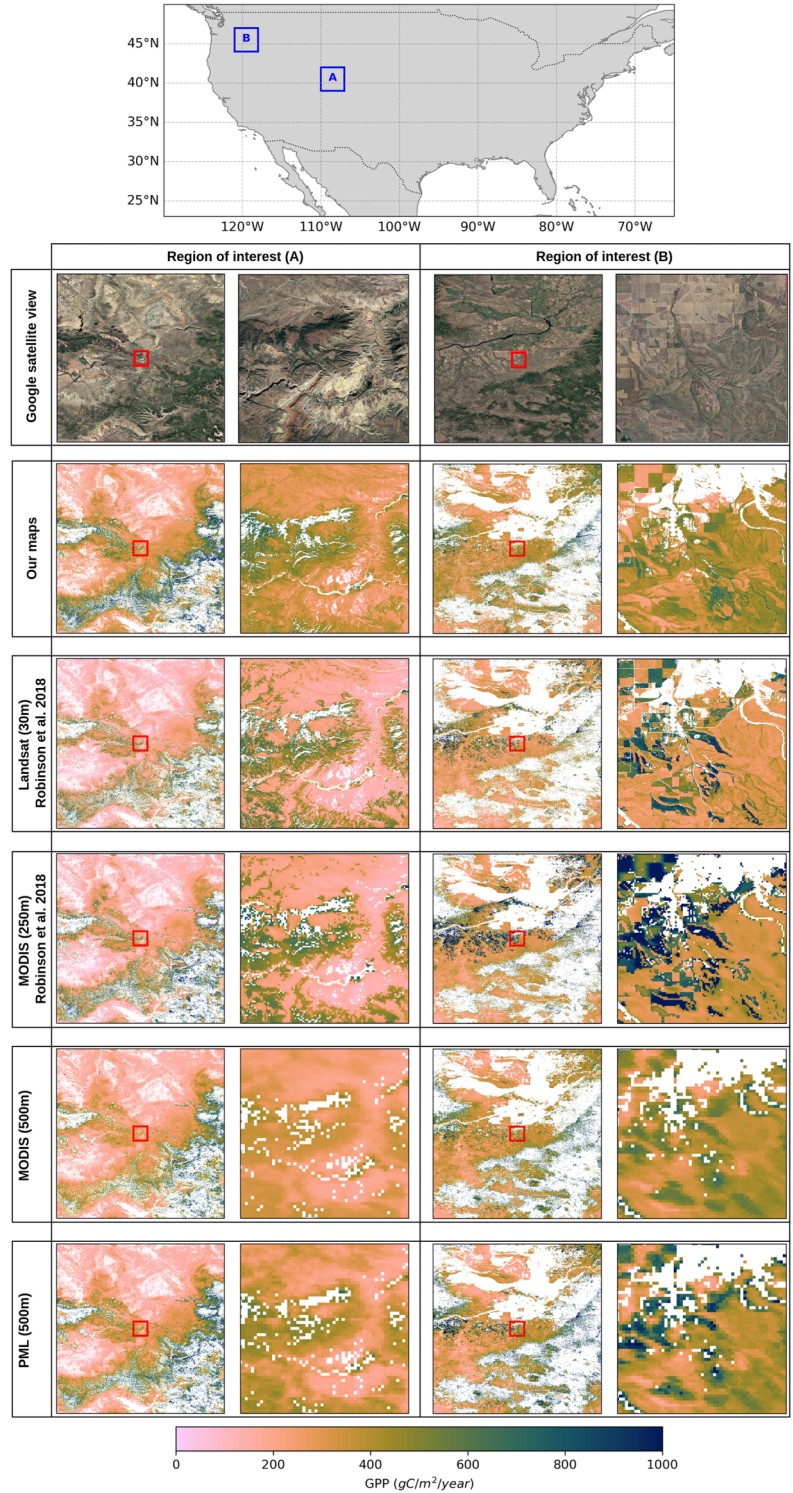

**Figure 8 Comparison of GPP products in grasslands over two regions of interest (A and B) in the CONUS.** The top map shows the locations of $3° \times 3°$ regions A and B with blue rectangles and the columns below present annual GPP maps in each region for 2018, with the left column showing region A and the right column showing region B. Red rectangles in the center of each GPP map correspond to $0.25° \times 0.25°$ close-up view in that region to emphasize the different resolutions. Satellite image: © 2025 Google.                 

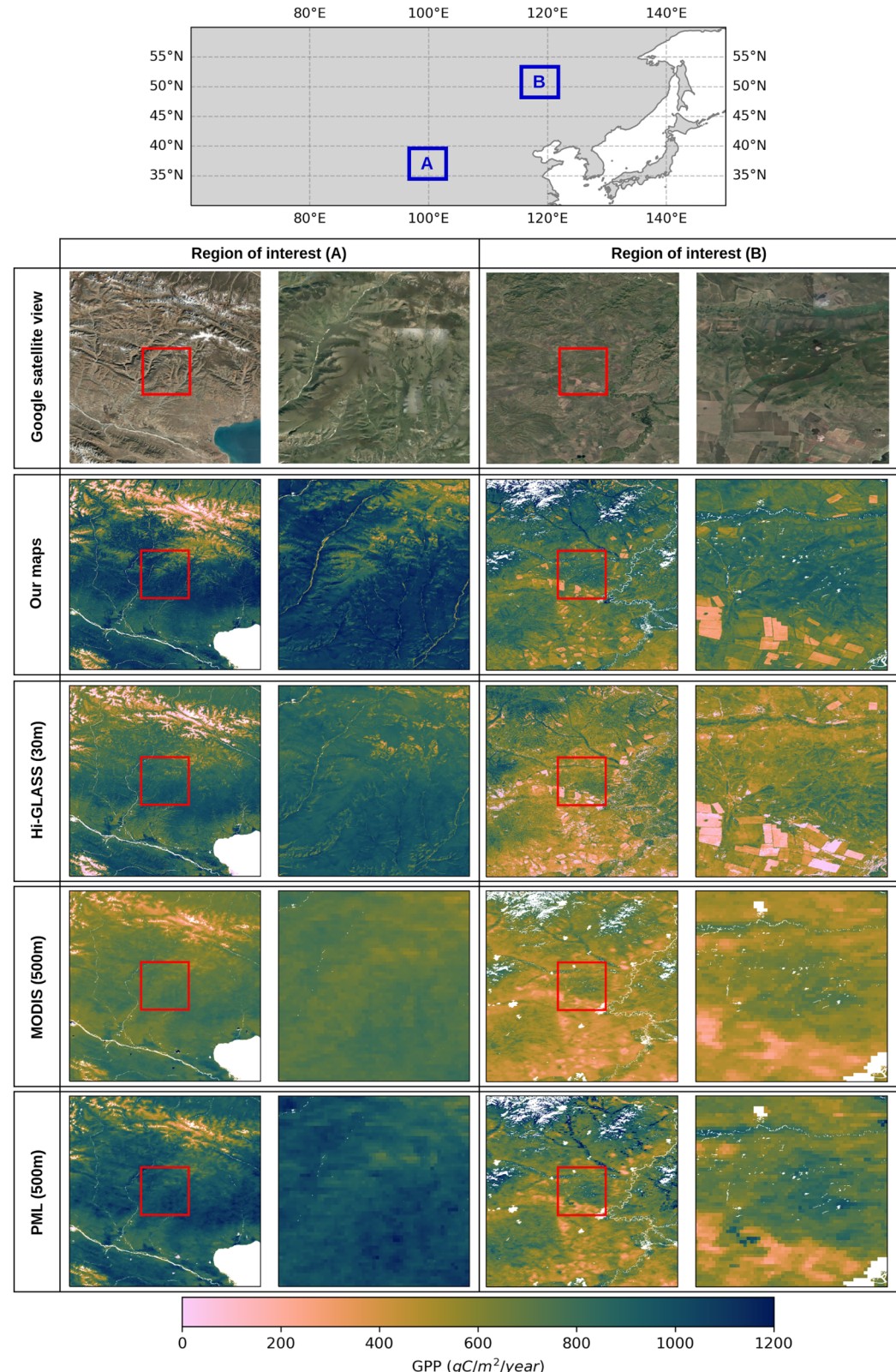

**Figure 9 Comparison of GPP products in grasslands over two regions of interest (A and B) in China.** The top map shows the locations regions A and B with blue rectangles and the columns below present

**Figure 9** (continued)
annual GPP maps in region A for 2016 and region B for 2020. Red rectangles in the center of each GPP map correspond to $0.25° \times 0.25°$ close-up view in that region to emphasize the different resolutions. Satellite image: © 2025 Google.               

Hi-GLASS, MODIS and PML models. Figure 9 illustrates a comparative evaluation of annual GPP estimates from four different remote sensing-based products—our model (30 m), Hi-GLASS (30 m), MODIS MOD17 (500 m), and the PML model (500 m)—across two regions in China where region A is in the Tibetan Plateau and region B is located in the north of China. The variations between our model, Hi-GLASS, and the existing coarser resolution GPP products (MODIS, PML) underscore the importance of spatial resolution in GPP estimation, particularly in regions with complex land cover configurations, as both 30 m resolution models effectively capture landscape heterogeneity and its influence on annual GPP estimates.

The two fine resolution maps depict the GPP spatial pattern in close agreement and capture the dynamics of annual productivity. Our model shows higher annual GPP values than both Hi-GLASS and MODIS, with patterns that align closely with land surface features. In both regions, PML model produces GPP values that are more comparable in magnitude to those from our model. However, the coarse resolution still limits its ability to resolve fine resolution productivity visible in both our model and Hi-GLASS. This highlights the added value of 30 m GPP products for regional-scale ecosystem monitoring. Among all four products, MODIS MOD17 shows the lowest GPP estimates which is consistent with previously reported underestimations of this product in grassland ecosystem in this region (*Zhu et al., 2018*). These biases are particularly evident in productive cropland areas and heterogeneous terrain, where the combination of coarse spatial resolution (500 m) and biome-averaged parameters results in excessive spatial smoothing and underrepresentation of $\varepsilon_{LUE_{max}}$ parameter for localized productivity peaks. Obviously, this underestimation cannot solely be attributed to the $\varepsilon_{LUE_{max}}$ parameter, as our model also adopts biome-specific $\varepsilon_{LUE_{max}}$ values derived from the MOD17 algorithm. Instead, part of the discrepancy stems from how environmental constraints on LUE, specifically temperature and water limitations, are implemented in the GPP modeling process, which can significantly influence spatial patterns and seasonal dynamics in GPP estimation.

## DISCUSSION

Grasslands are a natural habitat for a large number of species of plants and animals. This biodiversity increases the resilience of ecosystems and ensures ecological balance. Covering almost half of the Earth's surface, grasslands play a critical role in the global carbon cycle, as they are capable of storing considerable amounts of carbon and are an integral part of food supply and security (*Bai & Cotrufo, 2022*; *Liu et al., 2023*). For this reason, monitoring the productivity in grassland ecosystems at a high resolution over a long period of time is essential to control the level of degradation of these areas in the long term.

In this study, we produced 30 m resolution GPP products, both uGPP and GPP in grasslands, with global coverage between 2000 and 2022. These maps are produced in both bimonthly and annual periods, allowing the use of intra-annual GPP dynamics for observing short-term variations in grasslands and enabling robust computation of long-term GPP changes on an annual basis with a time series of over 20+ years. By analyzing annual fluctuations in GPP values, an extensive trend analysis can be performed to determine where the largest losses are, which could be an indicator of potential land degradation. In this way, it is possible to identify regions whose GPP is experiencing significant changes over time. Increasing GPP reflects improved productivity, which can be attributed to factors such as afforestation, improved agricultural practices, or favorable climate conditions. However, the decline in GPP in certain regions could indicate deforestation, land degradation, or drought impacts. These factors are crucial in assessing spatiotemporal variations in ecosystem health, carbon uptake dynamics, and climate scenarios (*Ding et al., 2021*; *Wang et al., 2021*; *Xie et al., 2020*).

## Accuracy of GPP estimations in grasslands

Accurate quantification of vegetation productivity contributes to a better estimate of carbon uptake, which is essential for understanding the spatial and temporal changes in the global carbon cycle. In our extensive comparisons with flux towers around the world, we have shown that our modeling approach is highly consistent with *in-situ* flux measurements. In the comparison of uncalibrated GPP products with flux towers, we observed high Pearson's correlation coefficients and $R^2$ values across different networks, which demonstrates the accuracy of our results (see Fig. 5).

Following the calibration of uGPP maps with $\varepsilon_{LUE_{max}}$ for grasslands from the MOD17 lookup-table (see Table 2), the grassland GPP maps showed a high correlation coefficient and a low bias in agreement with the *in-situ* observations. Despite a slight decrease in $R^2$ values in grassland ecosystems, compared to the validation in all biomes together, the GPP values can still account for more than 53% of the variation in GPP estimation at flux towers ($R^2 \sim 0.53 - 0.70$) with RMSE values smaller than 2.08 gCm$^{-2}$d$^{-1}$. The results agree with multiple previous studies that validated GPP estimations from the MOD17 algorithm in grassland ecosystems. For example, *Zhu et al. (2018)* validated the MODIS-based GPP estimations with flux towers from the Fluxnet2015 dataset. In this study, they only used flux towers in grassland regions and had five years or more time series of EC measurements to validate the model. By using monthly temporal periods, they found an agreement of $R^2 = 0.66$ and $RMSE = 2.46$ gCm$^{-2}$d$^{-1}$. *Sjöström et al. (2013)*, on the other hand, conducted a validation of the same model against the CarboAfrica flux network, focusing on grassland towers, and reported significantly poor results for MOD17 parameters. They showed that only 25% of the variations in GPP could be explained by the GPP model ($R^2 = 0.25$) when the MOD17 parameters were adopted, but it improved significantly up to $R^2 = 0.74$ when using site-specific values $\varepsilon_{LUE_{max}}$ for grasslands. *Pei et al. (2020)* assessed four GPP models (MOD17, PML, VPM, and BESS) against grassland flux towers from Fluxnet2015 data in the USA. The models showed a wide range of overall $R^2$ values ranging between 0.46 and 0.59 and RMSE values from 1.98 to 2.23 gCm$^{-2}$d$^{-1}$ where

MOD17 performed the worst ($R^2 = 0.46$, $RMSE = 2.23$ gCm$^{-2}$d$^{-1}$). *Robinson et al. (2018)* reported a similar RMSE (2.29–2.52 gCm$^{-2}$d$^{-1}$) for MOD17 in the same region, but using higher resolution remote sensing data. Similarly to *Sjöström et al. (2013)*, the statistics improved significantly when using the regionally calculated $\varepsilon_{LUE_{max}}$.

It is important to note that there are inherent uncertainties in the GPP estimates from the EC measurements themselves when comparing the modeled GPP *vs in-situ* data. Although flux towers offer useful information, some of the differences between model estimations and observations may stem from these uncertainties, which could account for minor variations. See *Tramontana et al. (2020)* for a detailed discussion on the limitations of DT and NT partitioning in GPP estimations.

## The effect of light use efficiency parameters

In light of studies that attempt to understand how well $\varepsilon_{LUE_{max}}$ represents the level of productivity for certain vegetation types, it is evident that MOD17 parameters suffer from significant problems of underestimation, most notably in cropland and forest areas, and overestimation in lower GPP values (*Wang et al., 2017*; *Yang et al., 2021*; *Zhang et al., 2012*). In an effort to increase the accuracy of GPP estimates regionally, several studies have tried to create ideal parameter values for particular biomes, and the results confirmed that most biomes had underestimated parameters compared to MOD17 parameters, despite overestimations in a few cases (*Madani et al., 2014*; *Zhang et al., 2008*). *Chen et al. (2021)* modeled the optimal value of $\varepsilon_{LUE_{max}}$, together with optimum growth temperature, globally as a spatial map through data-driven modeling, and this provides a potential approach to accounting for the spatial variation in vegetation productivity that extends the GPP and biomass studies.

The underestimation problem of MOD17 parameters is visible in our validations against flux towers in a broader comparison with all land cover classes together (see Fig. 5), though it is not the scope of this study to report this problem once more. In comparisons focused on grasslands, this is more clearly observed in the scattered plot of GPP values for flux towers and model estimates, as they diverge from the 1:1 line systematically. This can also be inferred from the constant negative biases presented in Fig. 5. This is also evident in previous studies in the literature (*Zhang et al., 2008*; *Zhu et al., 2018*). Furthermore, the comparison of our grassland GPP product with 30 and 250 m resolution GPP maps produced by *Robinson et al. (2018)* supported this argument, particularly in capturing regional dynamics across the CONUS region. As these GPP maps are adjusted using flux tower observations in the CONUS, they compensate for the underestimation of MOD17 parameters, hence resulting in higher, and more accurate, GPP estimations in this area. On the other hand, our grassland GPP maps show lower GPP estimations, while the spatial patterns match quite well.

Naturally, this discussion may raise the question: *"Why are we using MOD17 parameters to represent productivity in grassland regions in our final products when MOD17 is available at much coarser resolution than 30 m?"* Furthermore, we used these parameters to validate the results against flux towers for all land cover classes. Although this is indeed a valid point, it is precisely why we encode the $\varepsilon_{LUE_{max}}$ as a scaling factor in the

metadata of COG GeoTiffs, rather than directly applying it to the GPP values. In this setup, users can estimate and correct the scaling factor themselves, so that a more realistic GPP product can be produced locally. After long discussions and attempts to design the calculation steps aiming at highest accuracy and usability of data, we decided that this is the best we can do given the resources. Also note that we are presenting the uncalibrated version of GPP estimates by setting $\varepsilon_{LUE_{max}}$ as constant (the $\varepsilon_{LUE_{max}} = 1$ gCm$^{-2}$d$^{-1}$MJ$^{-1}$) everywhere, so that the productivity map can be free of any assumptions in terms of land cover class definitions and possible errors introduced by the (mis)classifications of the land cover map. Obviously, the problems in the estimation of GPP based on LUE cannot be solely attributed to the errors introduced by the land cover map. Even with a more accurate and detailed land cover map, $\varepsilon_{LUE_{max}}$ might still need an adjustment for catching local/regional dynamics in GPP (*de Almeida et al., 2018*).

Air temperature products are often coarse resolution compared to LST products (like MODIS, Landsat, or ECOSTRESS). It is even challenging to have a time-series of air temperature data spanning over two decades at finer spatial resolution. Globally available data sets such as ERA-5 are insufficient to capture high-resolution temperature stress as a result of their coarser resolution. Practically, the use of LST to estimate the temperature stress can be a more convenient approach than using the air temperature (*Karger et al., 2020*). For example, the MOD17 product uses a 1° resolution air temperature data set to determine the daily minimum temperature of 500 m pixel (*Running et al., 1999*). However, there are recent studies that demonstrated the benefits of using finer resolution LST data in GPP modeling. *Li et al. (2021)* used 70 m ECOSTRESS LST data to model GPP using a data-driven model. *Sims et al. (2008)* found LST more informative in the Temperature and Greenness (T-G) model, as it shows more about the canopy temperature than the air temperature.

## Limitations and known issues

The Landsat archive covers more than 20 years, allowing us to analyze ecosystem productivity in greater detail and capture finer changes compared to the MODIS archive. This wide temporal coverage, together with 30 m spatial resolution, enables an in-depth examination of spatiotemporal variations in productivity, seasonal fluctuations, and long-term accumulation of carbon stocks. However, a higher resolution dataset comes with several challenges that can limit the accuracy of GPP products in certain time frames and regions.

As remote sensing-based GPP models depend on the optical region of the electromagnetic spectrum, the quality and completeness of the input data sources are affected by the cloud covers and their shadows and other atmospheric and sensor artifacts. As these affected parts contain limited or no useful information, they are masked out from the satellite images, causing data gaps. The gap-filled version of the Landsat archive, however invaluable, still contains artifacts related to stripe effects or potential errors introduced by the time series reconstruction algorithm (see *Consoli et al. (2024*, "Validation at Flux Towers") for details), which causes alterations in the GPP estimations. In Fig. 10, we present the effect of stripes, due to the malfunctioning of the Landsat 7

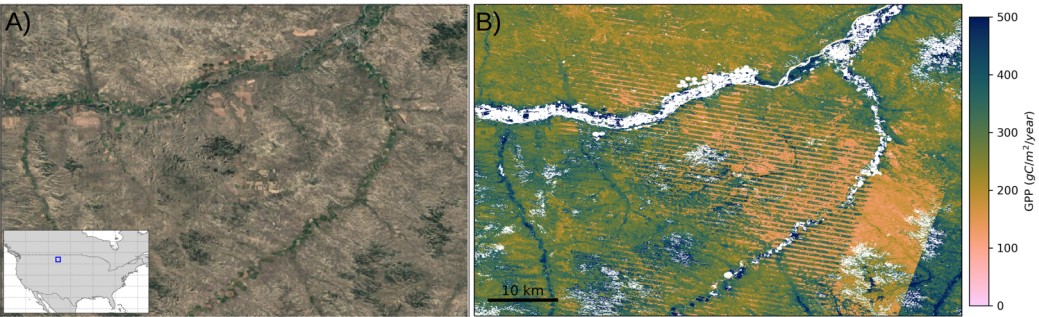

**Figure 10 An example stripe effect in grassland GPP map.** The example is shown from a region close to the northern border of the USA: Satellite view with the geographic location of the map (A) and annual GPP map (B) in the region for the year 2021. Satellite image: © 2025 Google.

sensor, in ARCO GeoTiffs on the GPP map. As can be clearly seen, the artifacts disrupt the spatial continuity of GPP estimations from the center of the image to the eastern side. Other sources of artifacts are the presence of snow and polar nights in winter months at high latitudes, which translates in missing images for consecutive months. This results in evident patterns from Landsat scenes that also impact the GPP maps.

Furthermore, bimonthly estimations of GPP may overlook growth periods in a plant's productivity or its rapid responses to rapid changes in environmental conditions. This can make it challenging for the model to estimate peaks appropriately in the time series and reflect seasonal productivity variations.

## Future work

As we process the Landsat ARCO archive continuously, we will further extend the temporal coverage of bimonthly estimates of uGPP and GPP in grasslands, which will eventually be aggregated to provide annual productivity layers. The continued operation of this product will obviously require the examination of other temperature estimates, such as Suomi-NPP VIIRS, Sentinel-3, and ECOSTRESS, since MODIS will soon cease operation (*Román et al., 2024*).

In the near future, we are planning to expand this computational frame to produce monthly aggregations of Landsat ARCO archives and explore the possibility of combining it with the Copernicus Sentinel-2 satellite of higher spatial and temporal resolution. This increase in the temporal resolution of GPP estimations will extend the possible use cases of these products to research areas where intraseasonal variations in vegetation productivity are important, such as agricultural monitoring. Using high temporal GPP values, it would be possible to monitor agricultural fields, especially on a field scale, by identifying crop phenology and agricultural periods, such as planting, growing, and harvesting seasons, which could be valuable for providing timely information on crop growth and the possible effect of environmental stressors (*Wang et al., 2023*).

Based on the regional assessment presented in Fig. 5, the model showed acceptable accuracy in regional networks with $R^2$ values around $\sim 0.5$–$0.6$. However, relatively lower

$R^2$ values, although still indicating high correlations, suggest a need for further adjustment to account for the regional dynamics of productivity. In order to address this issue, we aim to use recently available X-BASE products to calibrate GPP estimates for improved regional dynamics (*Nelson et al., 2024*). Given its spatial resolution, improved site coverage, and representation of diverse biomes and climatic conditions, X-BASE provides a strong reference for regional calibration.

The effect of water stress in vegetation, captured by the $W_{scalar}$ parameter, needs a deeper understanding at a finer resolution when derived from LSWI or vapor pressure deficit (VPD) as used in the MOD17 product. Moreover, the integration of Sentinel-2 to generate GPP at finer spatial and temporal resolutions will require further investigation to assess the effectiveness of $W_{scalar}$ derived from the NIR and green bands. For the temperature stress, on the other hand, despite the practicality of adopting a globally adjusted optimum temperature, we will explore the contributions of pixel-based optimum growth temperature in the calculation of $T_{scalar}$ (*Chen et al., 2021*).

We will also examine how precipitation, temperature, elevation, and anthropogenic activities affect productivity in grasslands, as well as how GPP dynamics relate to land degradation processes. We are currently conducting a separate study to systematically assess the trends in GPP and establish causal links with key environmental drivers and human-induced pressures. The goal of these studies is to improve our understanding of how grassland ecosystems function as well as to develop more effective monitoring and management strategies to detect and prevent land degradation.

## CONCLUSIONS

In this study, we present a novel framework for producing high-resolution global GPP maps for grassland regions from 2000 to 2022 in bimonthly and annual periods to monitor grassland ecosystem health globally under the Global Pasture Watch research consortium. By implementing a computational framework based on a light-use efficiency model using a gap-filled LANDSAT ARCO archive and MODIS-based temperature and CERES-based PAR products, we generated GPP maps globally at 30 m spatial resolution. These GPP products are, to the best of our knowledge, the first high-resolution GPP estimations produced on a global scale. The recommended uses of data include: trend analysis *e.g.*, to determine where the largest losses are and which could be an indicator of potential land degradation, crop yield mapping and for modeling GHG fluxes at fine spatial resolution.

Using an extensive eddy covariance flux tower data set (Fluxnet2015, Ameriflux, ICOS Warm Winter, Ozflux, and Chinaflux), we validated the performance of uGPP in all land cover classes together and the performance of grassland GPP separately. Although the uGPP model showed reasonably accurate results compared to flux towers, with significant improvements in the European region (ICOS network), the grassland GPP achieved considerably higher accuracy in the globally distributed Fluxnet2015 dataset. The models showed acceptable accuracies in other networks, but variations in accuracy suggest a need for further adjustment to better capture regional productivity dynamics.

We also provided access to the time-series of uncalibrated and grassland GPP maps as bimonthly (daily estimates in units of $gCm^{-2}d^{-1}$) and annual (daily average accumulated

by 365 days in units of $gCm^{-2}yr^{-1}$) products in Cloud-Optimized GeoTIFF ($\sim$23TB in size) as open data (CC-BY license). Users can access the bimonthly and annual maps through the SpatioTemporal Asset Catalog (http://stac.openlandmap.org) and the annual maps through the Google Earth Engine data catalog, along with the grassland probability maps that are used to produce annual grassland masks.

## ACKNOWLEDGEMENTS

The Landsat bands were made available, with many thanks to the University of Maryland GLAD lab. We gratefully acknowledge the contributions of the FLUXNET2015, ICOS, AmeriFlux, OzFlux, and ChinaFLUX networks for making the eddy covariance flux tower observations available.

### Funding

This work was supported by the Land & Carbon Lab from the Bezos Earth Fund. The Open-Earth-Monitor Cyberinfrastructure project has received funding from the European Union's Horizon Europe Research and Innovation Programme under grant agreement No. 101059548. The AI4SoilHealth project has received funding from the European Union's Horizon Europe research and innovation programme under grant agreement No. 101086179. The funders had no role in study design, data collection and analysis, decision to publish, or preparation of the manuscript.

### Grant Disclosures

The following grant information was disclosed by the authors:
Land & Carbon Lab from the Bezos Earth Fund.
European Union's Horizon Europe Research and Innovation Programme: 101059548 and 101086179.

### Competing Interests

Mustafa Serkan Isik, Leandro Parente, Davide Consoli and Tomislav Hengl are employed by OpenGeoHub. Lindsey Sloat and Radost Stanimirova are employed by World Resources Institute (WRI). Simone Sabbatini is employed by Euro-Mediterranean Center on Climate Change (CMCC). Nathaniel Robinson is employed by CIFOR-ICRAF. Ciniro Costa Junior is employed by Alliance of Bioversity International and CIAT.

### Author Contributions

- Mustafa Serkan Isik conceived and designed the experiments, performed the experiments, analyzed the data, prepared figures and/or tables, authored or reviewed drafts of the article, and approved the final draft.
- Leandro Parente conceived and designed the experiments, performed the experiments, analyzed the data, authored or reviewed drafts of the article, and approved the final draft.
- Davide Consoli conceived and designed the experiments, performed the experiments, analyzed the data, authored or reviewed drafts of the article, and approved the final draft.

- Lindsey Sloat conceived and designed the experiments, authored or reviewed drafts of the article, and approved the final draft.
- Vinicius Vieira Mesquita performed the experiments, authored or reviewed drafts of the article, and approved the final draft.
- Laerte Guimaraes Ferreira conceived and designed the experiments, authored or reviewed drafts of the article, and approved the final draft.
- Simone Sabbatini conceived and designed the experiments, authored or reviewed drafts of the article, and approved the final draft.
- Radost Stanimirova conceived and designed the experiments, authored or reviewed drafts of the article, and approved the final draft.
- Nathalia Monteiro Teles conceived and designed the experiments, authored or reviewed drafts of the article, and approved the final draft.
- Nathaniel Robinson conceived and designed the experiments, authored or reviewed drafts of the article, and approved the final draft.
- Ciniro Costa Junior conceived and designed the experiments, authored or reviewed drafts of the article, and approved the final draft.
- Tomislav Hengl conceived and designed the experiments, prepared figures and/or tables, authored or reviewed drafts of the article, and approved the final draft.

## Data Availability

The source code used to produce the GPP dataset is available at GitHub: https://github.com/wri/global-pasture-watch.

All GPP products are available at Zenodo: Isik, M. S. et al. (2025). Global Pasture Watch - Source Code of the Global Uncalibrated EO-based GPP and Grassland GPP Maps at 30m. Zenodo. https://doi.org/10.5281/zenodo.15675358.

The validation dataset is available at Zenodo: Isik, M. S., Parente, L., Consoli, D., Sloat, L., Mesquita, V., Ferreira, L., Sabbatini, S., Stanimirova, R., Teles, N., Robinson, N., Costa Junior, C., & Hengl, T. (2025). Global EC Flux Tower Bimonthly GPP Time Series from FluxNet2015, ICOS, AmeriFlux, OzFlux, and ChinaFlux for Product Validation [Data set]. Zenodo. https://doi.org/10.5281/zenodo.15349578.

Bimonthly and annual GPP products are available *via* SpatioTemporal Asset Catalog (STAC) of OpenLandMap:

- https://stac.openlandmap.org/gpw_ugpp.daily-30m/collection.json
- https://stac.openlandmap.org/gpw_ugpp-30m/collection.json
- https://stac.openlandmap.org/gpw_gpp.daily.grass-30m/collection.json
- https://stac.openlandmap.org/gpw_gpp.grass-30m/collection.json.

The annual GPP products are available at Google Earth Engine as part of Land & Carbon Lab Global Pasture Watch collections:

- projects/global-pasture-watch/assets/ggpp-30m/v1/ugpp_m
- projects/global-pasture-watch/assets/ggpp-30m/v1/gpp-grassland_m.

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
