# Peer review of "Light use efficiency (LUE) based bimonthly gross primary productivity (GPP) for global grasslands at 30 m spatial resolution (2000–2022)"

_PeerJ, doi:10.7717/peerj.19774_

## Round 0.1 · original submission · Major Revisions

Thank you for your paper submission. We have received comments from three reviewers, and all have highlighted the importance of your work, and I agree with this overall assessment. Two, however, have raised several points for clarification/ explanation. In particular, reviewer 2 has raised concerns regarding the experimental design and validation, and suggests an additional dataset. Reviewer 3 raises the same issue (point 7). Although you have assessed performance based on flux towers, please either address the concerns raised in your manuscript (i.e., perform additional verification), or explain in your review response if this isn't possible or beneficial. Further, I agree with reviewer 1 that it would be good to made the code used available, ideally as part of this review process.

Reviewer 1 ·

Basic reporting

The manuscript is well written.

Experimental design

The experimental design is logical, easy to follow, and presented in a way that is repeatable. It will be great if the authors can make the code accessible as well.

Validity of the findings

The findings are important and well evaluated.

Additional comments

The manuscript presented by Isik et al. addressed the long standing issue of the lack of a Landsat based GPP product. The paper is well written and presented. I have only minor comments.
I did not find any justification of why the maps are cropped for only grassland. I feel the original map of all terrestrial pixels would be of more interest. In addition, it is misleading to say the maps are evaluated against the over 500 flux towers in the abstract, when the actual number of flux towers in grasslands is lower?

Reviewer 2 ·

Basic reporting

This paper details the process of global GPP mapping, including the procedures and rich images. The research results have important contributions to promoting global GPP data sharing. Overall, this is an interesting study. However, there are still some suggestions.
1. In Figure 1, it is recommended to add arrows to indicate the relationship between annual grassland GPP and bimonthly grassland GPP.
2. The url link of Bimonthly uGPP is wrong.

Experimental design

1. Is the source of the annual grassland mask data reliable? It is crucial for estimating global grassland GPP. What are the connections and differences between global grassland extent data and annual grassland mask data?
2. Is it reasonable to set the optimal temperature for photosynthesis at 20.3℃ for different grassland types?

Validity of the findings

1. The authors used FLUX tower data for accuracy verification, but it seems to be lacking data from regions such as China and Mongolia. Considering the vast area of the Mongolian steppe, it is recommended to add some verification data.
The following websites may be helpful.
https://www.chinaflux.org/
https://doi.org/10.57760/sciencedb.o00119.00073
https://doi.org/10.57760/sciencedb.07138

Additional comments

no comment

Reviewer 3 ·

Basic reporting

no comment

Experimental design

no comment

Validity of the findings

no comment

Additional comments

In this study, the authors constructed a global 30-meter spatial resolution bimonthly gross primary productivity dataset for grasslands based on light use efficiency. The results of this study may contribute to monitoring global grassland ecosystem health and reveal the spatial and temporal changes characterizing their environment. However, there are some concerns that the authors should address before it can be considered for publication.
1. Lines 35-36, the authors should add more references to back "Grasslands cover almost 40% of the Earth’s land area, which makes them one of the most important ecosystems to maintain global ecological equilibrium (e.g., Shen et al., 2015, 2022; Ma et al., 2024)".
2. In the last paragraph of the introduction, the authors should highlight the significance of this study.
3. In Figure 1, I suggest to change the diagram name to flowchart and put the current diagram name in the method.
4. More mechanism explanations should be added to further explain how light use efficiency (LUE) can be used to produce gross primary productivity (GPP).
5. Is the research methodology of this study applicable to different ecosystems (e.g., forests, wetlands, etc.)?
6. How do the authors understand the impact of other environmental factors (e.g., precipitation, elevation, etc.) on grassland GPP?
7. Are the results of this study validated against existing high-resolution GPP data products from other regions (e.g., China) and how accurate are they?
8. In the conclusion, the authors should summarize the results and main findings of the paper in a condensed manner.

References:
Impact of climate change on temperate and alpine grasslands in China during 1982–2006. Advances in Meteorology, 2015, 2015(1): 180614.
Grassland greening impacts on global land surface temperature. Science of The Total Environment, 2022, 838: 155851.
Impacts of climate change on fractional vegetation coverage of temperate grasslands in China from 1982 to 2015. Journal of Environmental Management, 2024, 350: 119694.

---

## Round 0.2 · Minor Revisions

Thanks for making the changes requested in the first round of reviews. In my opinion the paper is now very close to ready, and it will form a very useful contribution. Before acceptance, please could you address the two minor points raised by reviewer 2, related to the number of flux towers and the accuracy metrics.

Reviewer 2 ·

Basic reporting

This manuscript has been revised, and I think it has been greatly improved. This is an interesting and meaningful study.

Experimental design

no comment

Validity of the findings

no comment

Additional comments

This manuscript has been revised, and I think it has been greatly improved. This is an interesting and meaningful study. Here are some further suggestions.
1. As reviewer 1 pointed out, there is some misunderstanding regarding the number of flux towers. Although in the revised version, the authors distinguished between all flux towers and flux towers in grassland areas, there is a specific number for all flux towers, but not for flux towers in grassland areas. I suggest also writing down the specific number of flux towers in grassland areas. Besides, why is it "more than 500"? I think it should have an exact number.
2. For Figure 5, I think the authors could put the accuracy metrics directly on the figure.

---

## Round 0.3 · accepted · Accept

Thank you for making the changes to your manuscript; I have checked the revisions and am now happy to recommend publication. Congratulations on your excellent work.